# Integration of single-cell transcriptomes and biological function reveals distinct behavioral patterns in bone marrow endothelium

Young-Woong Kim [1,9] ✉, Greta Zara[1], HyunJun Kang[1], Sergio Branciamore[2], Denis O'Meally [3], Yuxin Feng[4], Chia-Yi Kuan[5], Yingjun Luo [6], Michael S. Nelson [7], Alex B. Brummer[2,10], Russell Rockne [2], Zhen Bouman Chen [6,8], Yi Zheng[4], Angelo A. Cardoso[3,8] & Nadia Carlesso [1,8] ✉

Heterogeneity of endothelial cell (EC) populations reflects their diverse functions in maintaining tissue's homeostasis. However, their phenotypic, molecular, and functional properties are not entirely mapped. We use the *Tie2-CreERT2;Rosa26-tdTomato* reporter mouse to trace, profile, and cultivate primary ECs from different organs. As paradigm platform, we use this strategy to study bone marrow endothelial cells (BMECs). Single-cell mRNA sequencing of primary BMECs reveals that their diversity and native molecular signatures is transitorily preserved in an ex vivo culture that conserves key cell-to-cell microenvironment interactions. Macrophages sustain BMEC cellular diversity and expansion and preserve sinusoidal-like BMECs ex vivo. *Endomucin* expression discriminates BMECs in populations exhibiting mutually exclusive properties and distinct sinusoidal/arterial and tip/stalk signatures. In contrast to arterial-like, sinusoidal-like BMECs are short-lived, form 2D-networks, contribute to in vivo angiogenesis, and support hematopoietic stem/progenitor cells in vitro. This platform can be extended to other organs' ECs to decode mechanistic information and explore therapeutics.

As an interface between blood and tissues, the endothelium plays multiple roles in regulating tissues' homeostasis, response to stress and regeneration. These functional and regulatory properties involve highly heterogenous endothelial cell (EC) populations. Recent technological advances have greatly improved our understanding of EC diversity, showing molecular and functional heterogeneity of EC populations unique to each tissue[1,2]. However, comprehensive molecular characterizations and function attribution of tissue-specific EC populations are hampered by the inability to maintain primary EC in vitro without loss of their native identity and diversity.

The bone marrow (BM) remains one of the most challenging organs for EC isolation and characterization. Bone marrow endothelial cells

[1]Department of Stem Cell Biology and Regenerative Medicine, Gehr Family Center for Leukemia Research, Beckman Research Institute, City of Hope, Duarte, CA 91010, USA. [2]Department of Computational and Quantitative Medicine, Division of Mathematical Oncology, Beckman Research Institute, City of Hope, Duarte, CA 91010, USA. [3]Center for Gene Therapy, Beckman Research Institute, City of Hope, Duarte, CA 91010, USA. [4]Division of Experimental Hematology and Cancer Biology, Cincinnati Children's Hospital Medical Center, Cincinnati, OH 45229, USA. [5]Department of Neuroscience, Center for Brain Immunology and Glia (BIG), University of Virginia School of Medicine, Charlottesville, VA 22908, USA. [6]Department of Diabetes Complications and Metabolism, Beckman Research Institute, City of Hope, Duarte, CA 91010, USA. [7]Light Microscopy Core, Beckman Research Institute, City of Hope, Duarte, CA 91010, USA. [8]Irell and Manella Graduate School of Biological Sciences, Duarte, USA. [9]Present address: Center for Genome Engineering, Institute for Basic Science, Yuseong-gu Daejeon 34126, Republic of Korea. [10]Present address: Department of Physics and Astronomy, College of Charleston, Charleston, SC 29424, USA. ✉e-mail: ywkim@ibs.re.kr; ncarlesso@coh.org

(BMECs) are a critical component of the hematopoietic stem cell (HSC) niche during normal and malignant development[3]. Both arteriolar and sinusoidal BMECs play a pivotal, instructive role on HSC by regulating quiescence, self-renewal and trafficking[4–6]. BMECs are key for hemato-poietic regeneration following radiation and chemotherapy[7–9] and are active participants in malignant hematopoiesis[10–13]. Thus, a deeper understanding of BMECs, their diversity and crosstalk with HSC and other BM cell types, will lead to the identification of novel therapeutic targets. Attempts have been made to isolate and culture adult primary BMEC in vitro. Unfortunately, these efforts have been challenged by the difficulty of expanding these cells while maintaining their heterogeneity in culture.

To circumvent these problems a few approaches have been used. Investigators have been using other tissues sources of ECs to study their effects on HSC (e.g. mouse lung ECs or HUVECs)[14–16] or they have used either SV40 or myrAKT[17,18] to generate a uniform monolayer of immortalized EC. Recently, EC reprogramming by ETV2 reactivation opened a new avenue to decipher the crosstalk between organotypic ECs and parenchymal cells[19]. While these approaches are useful for in vitro expansion, they are limited in two main ways: the process of immortalization changes the nature of the EC, and reprogramming do not address directly the molecular and functional diversity and com-plex cellular interactions of primary BMEC.

We used a *Tie2-CreERT2;Rosa26-tdTomato* reporter mouse to trace, profile, and cultivate primary ECs from different organs. Here, we explored this approach to devise a method to dissect BMEC populations in vivo and in vitro. Based on previous efforts on human and murine EC isolation and assays[15,20,21] we hypothesized that isolation to purity may be detrimental for culture: BMECs likely need their native microenvironment, the interactions with other BM cell types and growth factors, that protect them from apoptosis and preserve their heterogeneity and functionality. Thus, we developed a traceable ex vivo EC culture system by using whole BM. This type of culture harbors two different types of BMEC distinguishable by their migra-tory behavior and by the expression of *Endomucin* (Emcn), which in vivo marks venous/sinusoidal ECs[22,23]. These two populations exhibit distinct arterial-like and sinusoidal-like molecular signatures and unanticipated functional characteristics. Single-cell RNA-sequencing (scRNA-seq) results demonstrate that the in vivo transcriptomic pro-files of BMECs are maintained ex vivo for a period of two weeks. Taken together, we have established a surrogate method for studying BMEC functions, which can serve as a paradigm for other tissue-specific ECs. This platform can be used to conduct mechanistic studies and for drug screenings with either anti-angiogenic or pro-angiogenic molecules in the context of developing therapies.

## Results

### Marking of adult tissues' vasculature in vivo and in vitro

The Tie2 receptor (*Tek* gene), essential for the remodeling and maturation of blood vessels, is expressed primarily in ECs[24]. We used a recently developed Tamoxifen inducible *Tie2-CreERT2* transgenic mouse[25] crossbred with the *Rosa26-tdTomato* reporter mouse to track ECs in vivo and in vitro. Following Tamoxifen induction, the activity of the *Tek* promoter-driven Cre recombinase was examined in bone, spleen, kidney, heart, lung, and liver. Confocal imaging of tissue sec-tions showed that tdTomato (tdT) expression was strictly confined to vascular structures expressing the endothelial marker CD31 (Fig. 1a). Flow cytometry analysis confirmed that tdT+ cells were all negative for the hematopoietic marker CD45 and positive for the endothelial marker CD31 in all organs examined (Fig. 1b). There was variability in the representation of tdT+ cells in the various tissues: from 0.007% in the spleen to 6.028% in the heart (Supplementary Fig. 1a). In the bone, CD45⁻Ter119⁻tdT+ cells represented about 0.012% of total BM (range 0.0015–0.020%; Supplementary Table 1). tdT+ cells were detected only in the CD45⁻ fraction (Supplementary Fig. 1b), and were positive for the

endothelial markers CD31, CD144, Flk-1, Sca-1, and CD105; partly positive for the endothelial marker Emcn, and negative for the myeloid cell marker CD11b (Fig. 1c). Next, we harvested BM, spleen, kidney, heart, lung, liver, and SAT (subcutaneous adipose tissues), and cul-tured each whole cell suspension on collagen plates for two weeks. Tissues specific tdT+ ECs were easily distinguishable in culture, show-ing distinct morphology and growth patterns in the different tissue cultures (Supplementary Fig. 1c). These results indicate that the *Tie2-CreERT2;Rosa26-tdTomato* (herein *Tie2Cre^{ER}tdT*) mouse model marks specifically ECs and is an effective tool for tracing and genetically manipulating mouse ECs in vivo and in vitro.

### BMEC heterogeneity requires non-cell autonomous cues

While protocols for murine EC isolation and culture from various organs (kidney, heart, lung, liver, brain) have been standardized[1,2,26], isolation and culture of primary BMEC remain a challenge, hindering the efforts to decode mechanistic information[27]. We used the *Tie2Cre^{ER}tdT* mouse model to develop a more effective approach for BM. First, we sorted BMECs directly from fresh bones (T0^{tdT+} BMEC) and cultured them in EC-specific media. Sorted T0^{tdT+} BMECs formed uniform monolayers that became confluent in two weeks (Fig. 2a and Supplementary Fig. 1d). In culture, they expanded on average 7-fold by day 14 (up to $8.4 \times 10^4$ total cells, Supplementary Table 2) but were not able to form structures in Matrigel tube formation assay (Fig. 2c). These data suggest that culture of sorted, pure BMECs is suboptimal and results in a uniform culture with limited expansion and poor ability to engage in angiogenesis in vitro.

We hypothesized that in order to preserve their functionality, BMECs need conditions closer to their native in vivo microenviron-ment, such as cell to cell interactions with other BM cell types and soluble factors. Therefore, we established cultures of whole BM (WBM), where BMECs could be traced within the other BM cells by tdT expression. WBM freshly isolated from *Tie2Cre^{ER}tdT* mice were plated and cultured in the presence of EC-specific media for two weeks. This initial culture (day 0-14) is herein defined as Passage 0 WBM (P0 WBM) and referred as ex vivo. Cultures of sorted or serially passaged cells beyond P0 are here referred as in vitro.

Time lapse microscopy of P0 WBM cultures showed that tdT+ BMECs (P0^{tdT+}) single cells (Supplementary Fig. 1e) evolved to form two types of structures by day 4–5 after seeding (Fig. 2a, right panels, Supplementary Fig. 1f): some tdT+ cells formed two-dimension (2D)-web-like networks (Supplementary Fig. 1f, g), others formed colonies (Supplementary Fig. 1f, h). We termed them "2D-network forming" and "colony forming" BMECs, respectively (Supplementary Movie 1 and 2). The 2D-network structures produced by BMECs in P0 WBM cultures were never observed in cultures from other tissues (Supplementary Fig. 1c) or in cultures of sorted BMEC T0^{tdT+} (Fig. 2a). In striking contrast with sorted T0^{tdT+} BMECs, P0^{tdT+} BMEC expanded an average 260-fold during the two weeks culture (up to $3.4 \times 10^6$ total cells; Fig. 2b, Sup-plementary Table 3) and formed a network of structures defined here as "cords" (cord-networks) when seeded in Matrigel tube assay (Fig. 2c, Supplementary Movie 3, and Supplementary Fig. 5c, d). Note that this type of structures are commonly referred as "tubes", even in the absence of lumen[28–31]. Alike fresh tdT+ cells, cultured P0^{tdT+} cells exhib-ited expression of endothelial markers CD31, CD144, Flk-1, and Emcn, as well as of Sca-1 and CD105 (Fig. 2d). Expression of Emcn allowed to clearly distinguish an Emcn negative population and an Emcn positive population. In summary, ex vivo WBM culture promotes a remarkable expansion of BMECs and preserves heterogeneous BMEC populations with distinct migratory behaviors.

### Emcn expression identifies functionally distinct BMEC types

Emcn expression defines in vivo venous/sinusoidal ECs and is absent in arterial ECs[22,23]. We observed a similar pattern in the *Tie2Cre^{ER}tdT* reporter mice: Emcn was expressed in the microvasculature of the bone and of other organs, but not in the arteries, which were identified also

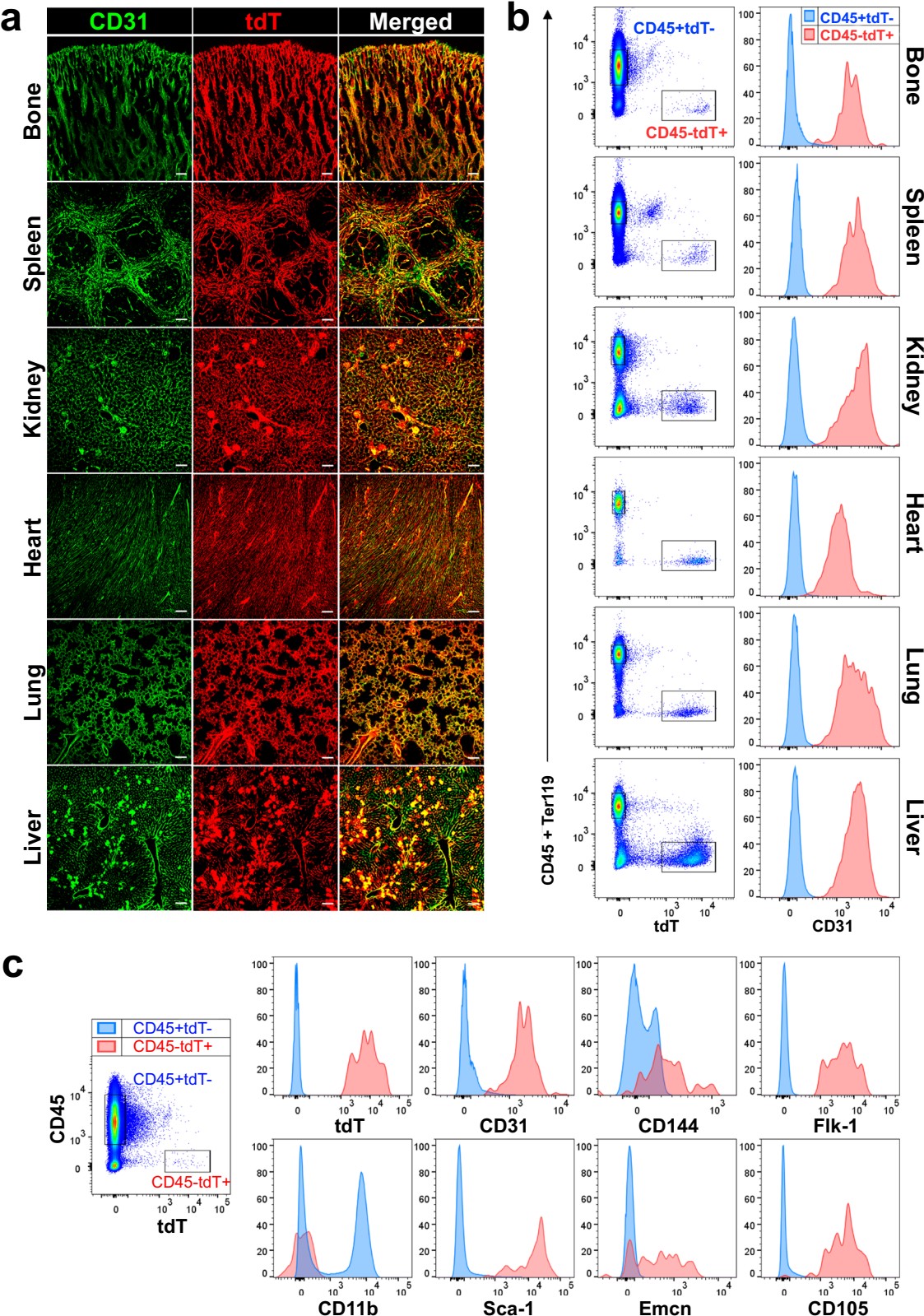

**Fig. 1 | Endothelial cell-specific tdT reporter expression in *Tie2-CreERT2;Rosa26-tdTomato* mice. a** Confocal images of CD31 (in green) staining and tdT reporter expression (in red) on the indicated tissues. Scale bar, 100 μm. Representative of *n* = 2 independent experiments. **b** Flow cytometry analysis of the dissociated cells from the indicated tissues for tdT (**b**, left) and CD31 expression (**b**, right).

Representative of *n* = 5 independent experiments. **c** Expression of the indicated EC markers on fresh BM tdT⁺ cells from 4-6-month-old *Tie2-CreERT2;Rosa26-tdTomato* mice. Histograms show the expression of each marker in CD45⁺ tdT⁻ hematopoietic cells (blue) and CD45⁻tdT⁺ ECs (red). Representative of *n* = 5 independent experiments. Gating strategies are shown in Supplementary Fig. 9.

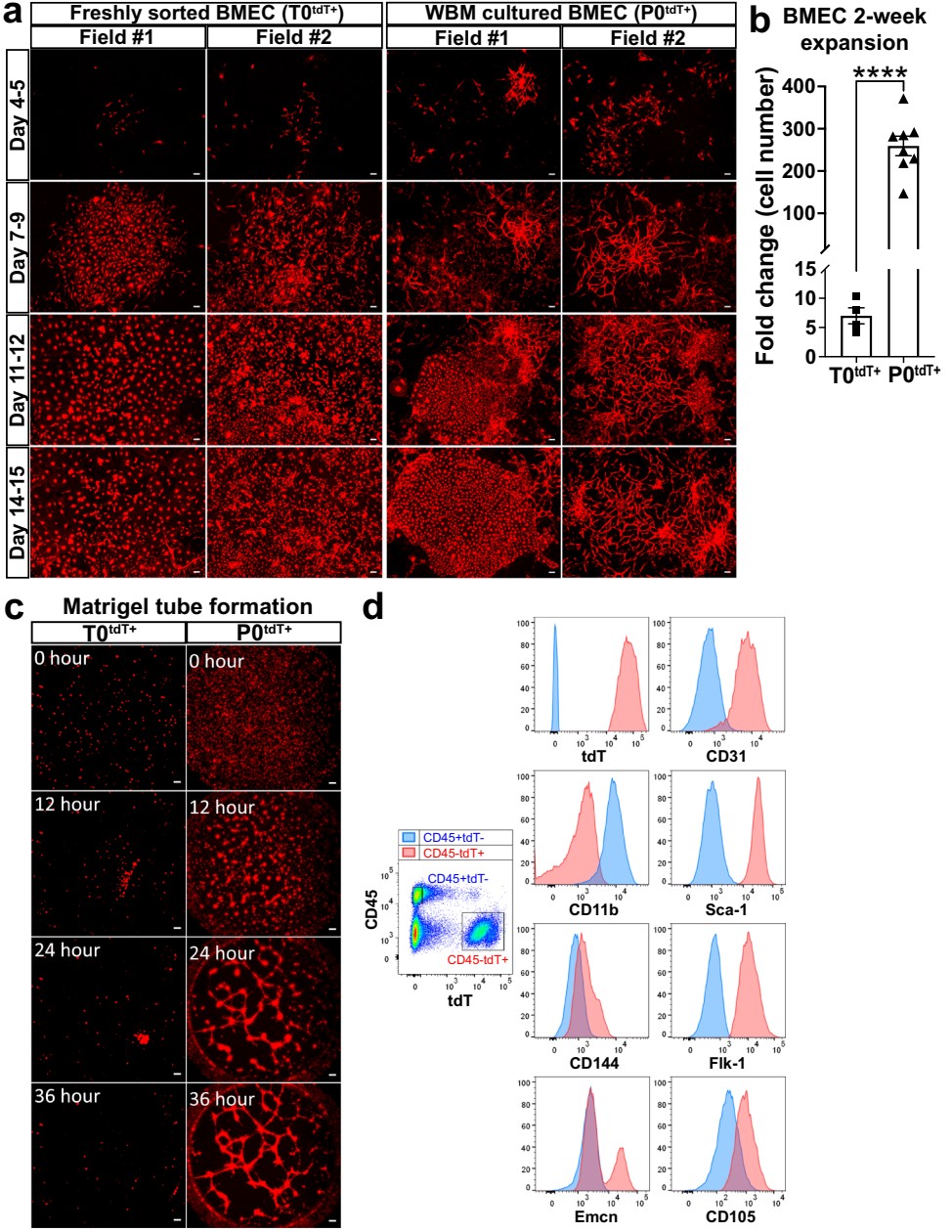

**Fig. 2 | Ex vivo WBM culture system preserves BMECs capable to form 2D-networks. a** Snapshot images of two representative fields of tdT⁺ BMEC cultures directly sorted from fresh BM (T0tdT⁺; left) or WBM culture (P0tdT⁺; right) at the indicated day of culture. Scale bar, 100 μm. Representative of $n = 4$ independent experiments. **b** Fold increase of absolute numbers of CD45⁻tdT⁺ cells from each culture (day 14/day 0). T0tdT⁺ ($n = 4$), P0tdT⁺ ($n = 8$). Data are presented as the mean ± SEM; statistics were determined using unpaired t-test, two-sided, $p = 0.0001$. Source data are provided as a Source Data file. **c** Time lapse images of Matrigel tube formation assay with CD45⁻Ter119⁻tdT⁺ cells directly sorted from BM (T0tdT⁺; left), or with CD45⁻tdT⁺ cells sorted after two weeks of P0 WBM culture (P0tdT⁺; right). Scale bar, 200 μm. Representative of $n = 4$ independent experiments. **d** BMECs from P0 WBM culture at day 14. Histograms show expression of the indicated EC markers on gated CD45⁺tdT⁻ hematopoietic cells (blue) and CD45⁻tdT⁺ ECs (red). Representative of $n = 4$ independent experiments. Gating strategies are shown in Supplementary Fig. 9.

by expression of αSMA (Supplementary Fig. 2a–c). Immunostaining of fixed P0 WBM cultures showed that both 2D-network-forming and colony-forming BMECs expressed CD144/VE-Cadherin and CD31 (Fig. 3a, b) but that only the 2D-network-forming BMECs expressed Emcn (Fig. 3c and Supplementary Fig. 2d). Flow cytometry analysis showed that a distinct Emcn⁺ population, present in fresh BM and in all organs, was preserved mainly in P0 WBM day 14 cultures (Fig. 3d and Supplementary Fig. 2e, f). Sca-1 expression is commonly used to distinguish arterial vs. venous/sinusoidal ECs: Sca-1ʰⁱ ECs are defined as arterial-like ECs whereas Sca-1ˡᵒ ECs are considered venous/sinusoidal-like[32]. We found that the colony-forming Emcn⁻ cells were Sca-1ʰⁱ

whereas the network-forming Emcn⁺ cells were Sca-1ˡᵒ (right and bottom left panels of Fig. 3d). Thus, according to this classification, the 2D-network-forming cells exhibit a venous/sinusoidal-like phenotype and the colony-forming cells an arterial-like phenotype.

In P0 WBM culture at day 14, total tdT⁺ BMECs comprised ~20% of the entire culture (Figs. 3e and 4a). Such tdT⁺ population contained on average 15% 2D-network-forming cells Emcn⁺Sca-1ˡᵒ BMECs (venous/sinusoidal-like) and approximately 85% of colony-forming cells Emcn⁻Sca-1ʰⁱ BMECs (arterial-like; Fig. 3d, f). Over time, Emcn⁺Sca-1ˡᵒ BMECs decreased from 15% at P0 to ~2% at P1 and were barely detected at P2. In contrast, Emcn⁻Sca-1ʰⁱ BMECs increased to ~100% of

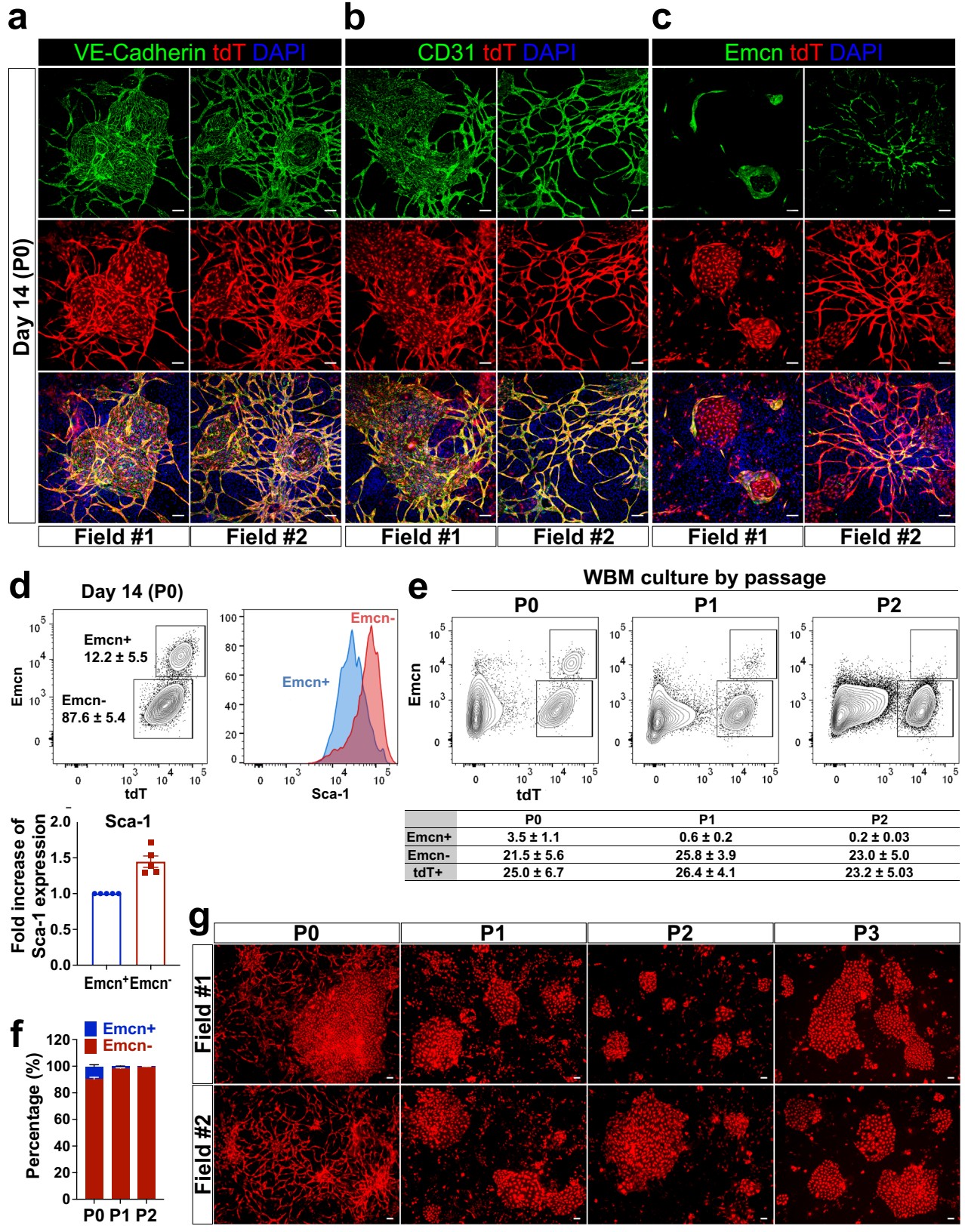

BMECs at the end of P2 (Fig. 3e, f). In culture, 2D-networks formed by Emcn$^+$Sca-1$^{lo}$ BMECs were no longer detected at P1 and at later passages (Fig. 3g).

These results demonstrate that the growth of both Emcn$^+$Sca-1$^{lo}$ venous/sinusoidal-like and Emcn-Sca-1$^{hi}$ arterial-like BMECs is preserved and supported by the WBM culture for the first two weeks.

**WBM populations and their kinetics in long-term culture**

Our results suggest that the presence of other cell lineages in culture is critical for preserving BMEC heterogeneity and function. Flow cytometry analysis of fresh BM at day 0 and of P0 WBM culture at day 14 revealed three major populations (Fig. 4a): (i) CD45$^-$tdT$^+$ (BMECs); ii) CD45$^-$tdT$^-$ mesenchymal cells (MCs), and (iii) CD45$^+$tdT$^-$ hematopoietic

**Fig. 3 | Endomucin expression defines two distinct BMEC populations in WBM culture. a–c** P0 WBM culture at day 14. Immunostaining of tdT⁺BMECs with antibodies for VE-Cadherin (**a**), CD31 (**b**), and Emcn (**c**). Confocal images of two fields in the two independent cultures. Scale bar, 100 μm. Representative of $n = 3$ independent experiments. **d** Emcn expression on gated tdT⁺BMECs in P0 WBM culture at day 14. Emcn⁺ and Emcn⁻ populations (dot plot, left) were gated and analyzed for Sca-1 expression (histogram, right). Representative of $n = 5$ independent experiments. Bar graph (bottom left) summarizes fold increase in mean fluorescence intensity of Sca-1 expression in Emcn + (blue) vs. Emcn- (red) cells; $n = 5$. Data are presented as the mean ± SEM. Gating strategies are shown in Supplementary

Figure 9. Source data are provided as a Source Data file. **e, f** At day 14, P0 WBM cultures were serially passaged every seven days and analyzed by flow cytometry at each passage. **e** Dot plots show Emcn⁺ and Emcn⁻ cells on gated tdT⁺ BMEC from P0 to P2 in a representative of $n = 3$ independent experiments. Source data are provided as a Source Data file. **f** Bar graph summarizes the relative proportion of the Emcn⁺ vs. Emcn⁻ cells at P0–P2. P0 $n = 10$; P1 $n = 8$; P2 $n = 8$. Source data are provided as a Source Data file. **g** At day 14, the P0 WBM culture was serially passaged every 7–14 days (P0 to P3). Confocal images of two fields at the last day of the indicated passage. Scale bar, 100 μm. Representative of $n = 5$ experiments. Note that web-like networks (bottom left) were no longer present after P0.

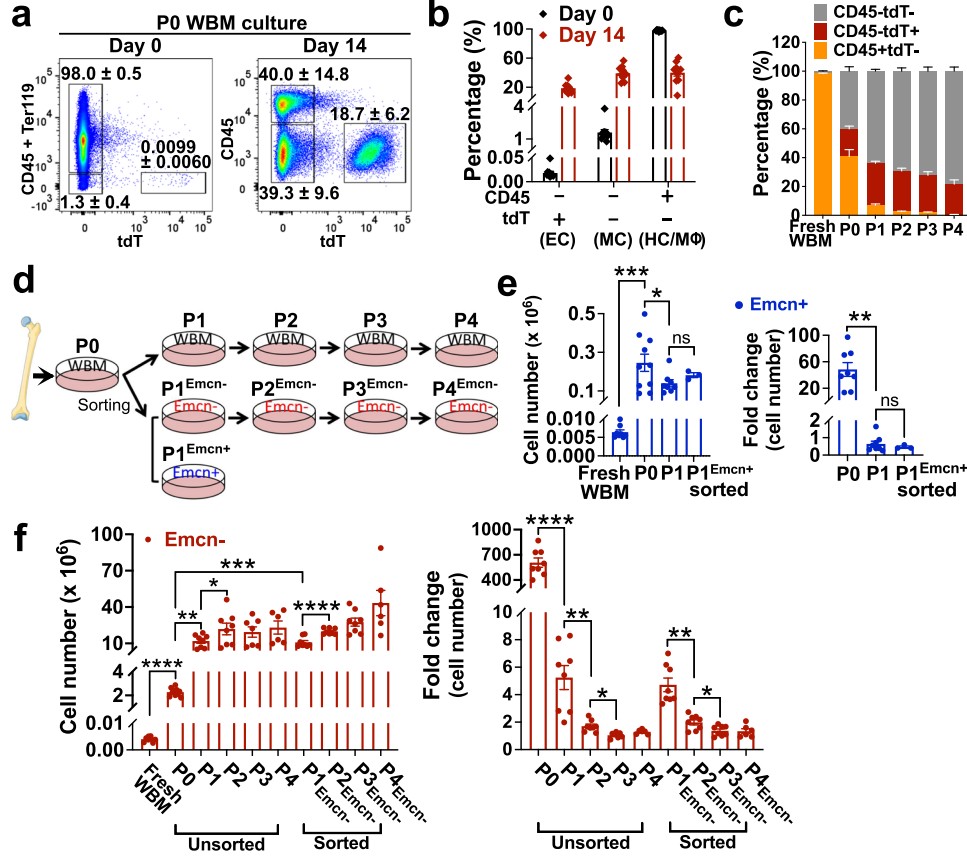

**Fig. 4 | Characterization of WBM culture populations over time. a** Dot plots within gates show distribution of three different populations, CD45⁺tdT⁻ (HC/Mφ), CD45⁻tdT⁻ (MC), and CD45⁻tdT⁺(EC), at day 0 and 14 of P0 WBM culture. Representative of independent experiments: $n = 14$ at day 0 and $n = 10$ at day 14. Source data are provided as a Source Data file. **b** Percentages of populations at day 0 (black bar) and 14 (red bar) of P0 WBM culture; $n = 10$. Data are presented as the mean ± SEM. Source data are provided as a Source Data file. **c** P0 WBM cultures were harvested and serially passaged up to 4 times (P4). At each passage, the relative percentages of CD45⁻tdT⁺ (BMEC, red), CD45⁻tdT⁻ (MC, gray), and CD45⁺tdT⁻ (HC/ Mφ, CD11b⁺, orange) populations were evaluated by flow cytometry; Fresh WBM $n = 10$; P0 $n = 10$; P1 $n = 8$; P2 $n = 8$; P3 $n = 7$; P4 $n = 6$. Data are presented as the mean ± SEM. Source data are provided as a Source Data file.(**d**) Scheme of experimental design. Parts of the figure (the femur) were drawn by using pictures from Servier Medical Art by Servier, licensed under a Creative Commons Attribution 3.0 Unported License. At P0, tdT⁺ BMECs were: (i) passaged when confluent (every week) as WBM cultures; (ii) sorted into Emcn⁻ and Emcn⁺, plated as individual cultures and passaged once confluent (every week). **e** Total cell number

of Emcn⁺ cells counted at the indicated passage (left); fold changes of cell numbers at the indicated passage compared to the previous one (right). Fresh WBM $n = 8$; P0 $n = 10$; P1 $n = 8$; sorted P1^Emcn⁺ $n = 3$. Data are presented as the mean ± SEM. Statistics for all comparisons shown were determined using paired t-test, two-sided. Left graph: Fresh WBM vs. P0, $p = 0.0008$; P0 vs. P1, $p = 0.0193$. Right graph: P0 vs. P1, $p = 0.0027$. * $p < 0.05$, **$p < 0.01$, ***$p < 0.001$. Source data are provided as a Source Data file. **f** Total cell number of Emcn⁻ cells counted at the indicated passage (left); fold changes of cell numbers at the indicated passage compared to the previous one (right). Fresh WBM $n = 8$; P0 $n = 10$; unsorted P1 and P2, sorted P1^Emcn⁻, P2^Emcn⁻, and P3^Emcn⁻ $n = 8$ (each group); unsorted P3 $n = 7$; unsorted P4 $n = 6$, sorted P4^Emcn⁻ $n = 6$. Data are presented as the mean ± SEM. Statistics for all comparisons shown were determined using paired t-test, two-sided. Left graph: Fresh WBM vs. P0, $p < 0.0001$; P0 vs. P1, $p = 0.0015$; P1 vs. P2, $p = 0.0151$; P0 vs. P1^Emcn⁻, $p = 0.0005$; P1^Emcn⁻ vs. P2^Emcn⁻, $p < 0.0001$. Right graph: P0 vs. P1, $p < 0.0001$; P1 vs. P2, $p = 0.0027$; P2 vs. P3, $p = 0.0141$; P1^Emcn⁻ vs. P2^Emcn⁻, $p = 0.0045$; P2^Emcn⁻ vs. P3^Emcn⁻, $p = 0.0296$. *$p < 0.05$, **$p < 0.01$, ***$p < 0.001$, and ****$p < 0.0001$. Source data are provided as a Source Data file.

cells/macrophages (HC/Mφ). In the 2-weeks WBM culture, BMECs were enriched from ~0.01% to ~20% and MCs were enriched from ~1% to ~40%, whereas HC/Mφ decreased from 98% to ~40% (Fig. 4b). Lineage identity of these cell populations was confirmed by gene expression analysis of EC, HC/Mφ, and MC markers (Supplementary Fig. 3a).

Further immunophenotypic analysis showed that, at day 0, the HC fraction comprised CD3⁺ T-cells, B220⁺ B-cells, and CD11b⁺ myeloid cells; by day 5 of culture and beyond, CD3⁺ and B220⁺ cells were barely detected, and by day 14, all CD45⁺tdT⁻ HC cells were constituted solely by adherent macrophages (Mφ) expressing the myeloid/macrophage

markers CD14, CD11b, and F4/80 (Supplementary Fig. 3b, c). The stromal population CD45⁻tdT⁻ expressed the MC markers PDGFRα (CD140a), CD90, CD44, CD29, and Sca-1 (Supplementary Fig. 3c).

After P0, WBM cultures could be serially passaged for several times. BMECs had their greatest expansion within the first two weeks of WBM culture (P0) and proliferated up to Passage 4 (P4), constituting a steady 30% of the entire WBM population from P1 to P4 (Fig. 4c and Supplementary Fig. 3d). MΦ decreased significantly from P0 to P1 (from 40% to 5·7%) and throughout serial passages (to ~2%; Fig. 4c and Supplementary Fig. 3e) suggesting limited expansion in this culture system. Unlike MΦ, MCs expanded during the serial passages and dominated the WBM culture throughout P1-P4 representing ~75% of the culture at P4 (~62-fold total expansion at P4; Fig. 4c and Supplementary Fig. 3f).

## Emcn⁻ BMECs are endowed with high proliferative potential

We characterized the proliferative potential of Emcn⁺ and Emcn⁻ BMECs and assessed how their growth depended on WBM culture after P0. P0 WBM cultures were harvested at day 14 and BMECs were passaged using two approaches: (1) BMECs were replated and serially passaged as WBM; (2) BMECs were sorted into Emcn⁺ and Emcn⁻ fractions and each population was separately replated and serially passaged as homogeneous populations (Scheme in Fig. 4d).

The greatest expansion of BMEC (both Emcn⁺ and Emcn⁻) occurred within the first two weeks of WBM culture (P0) (Fig. 4e, f). The absolute number of Emcn⁺ BMECs increased significantly (48-folds; Supplementary Table 4) during P0, but they rapidly declined at P1 and were no longer detected at P2, regardless of whether Emcn⁺ cells were growing as WBM culture (Figs. 3e and 4e) or as sorted cells (Fig. 4e). Emcn⁺ cells formed 2D-networks only at P0 (Supplementary Fig. 3g). Of note, the exhaustion of Emcn⁺ cells in WBM from P0 to P2, significantly correlated with the decline of MΦ (Pearson $r = 0.997$; $p = 0.0002$; Fig. 4c, e).

Emcn⁻ cells increased greatly (~600-fold; Supplementary Table 4) during P0 culture, and unlike the Emcn⁺ cells, they propagated in culture for up to four passages (Fig. 4f). Although no statistically significant differences were observed between the proliferative rates of Emcn⁻ BMECs passaged as WBM culture or as sorted cells, sorted Emcn⁻ cells showed a trend for greater expansion from P1 to P4 (~20-fold) compared to Emcn⁻ cells passaged as WBM (~10-fold; Fig. 4f and Supplementary Fig. 3h). We observed a negative correlation between MC and Emcn⁻ cells in WBM P1-P4 (Pearson $r = -0.944$; $p = 0.055$; Fig. 4c, f); thus, it is likely that the greater expansion of sorted Emcn⁻ cells is due to absence of an inhibitory effect exerted by MCs. The rates of expansion of Emcn⁻ BMECs were high from P0 to P1 (~5-fold) but gradually decreased during subsequent passages (Fig. 4f, right panel. Emcn⁻ BMECs proliferated as a cobblestone monolayer from P1 to P4 (Supplementary Fig. 3g) and could generate up to either ~23 million (WBM) or ~32 million (sorted) BMECs from an initial culture of ~10,000 cells/mouse. Altogether, the overall expansion of total CD45⁻tdT⁺ BMECs from day 0 to P4 of WBM culture was greater than 5000-fold.

## MΦ promote preservation of Emcn⁺ and expansion of Emcn⁻ BMECs

The above studies provided us with three key results: (1) MΦ decreased significantly from P0 to P1; (2) MC increased over time; and (3) Emcn⁺ BMECs became undetectable after P1 (Figs. 3e–g and 4c, e). The significant correlation between Emcn⁺ BMEC depletion, decrease in MΦ numbers and expansion of MC prompted us to explore whether changes in populations' ratio impacted BMECs output. Treatment of P0 WBM with additional MΦ significantly increased the total number of tdT⁺ BMECs vs. control condition: twice as many MΦ added to standard WBM culture resulted in a 2-fold further increase of P0 Emcn⁺ over the T0 baseline (~96-fold vs. 48-fold), particularly at lower cell densities (Fig. 5a, c, d). A similar

enhancement was seen for P0 Emcn⁻, and for all BMEC, as quantified by confocal microscopy (Supplementary Fig. 4a). Coculture using transwell inserts suggested that the effect of MΦ is mediated mostly by soluble factors (Supplementary Fig. 4c).

In contrast, supplement of additional MCs was associated with a decrease in the total number of BMECs, both Emcn⁻ and Emcn⁺ BMECs (Fig. 5b–d). Coculture with transwells suggested that the effect of MC is mediated mostly by cell-to-cell contact (Supplementary Fig. 4c).

Given these results, we hypothesized that supplementary BM MΦ could rescue the cultures of sorted BMECs (T0^tdT⁺; Fig. 2a) and/or the serial passages of WBM (Fig. 3e–g), where the absence or decline in MΦ numbers correlated with the absence of Emcn⁺ cells. BMECs directly sorted from fresh bone were examined upon treatment with supplementary MΦ, supplementary MCs, or a combination of both. Addition of MΦ resulted in a greater expansion of sorted T0^tdT⁺ BMECs compared to sorted T0^tdT⁺ BMECs cultured alone, and rescued T0^tdT⁺ from the inhibitory effect of MCs (Supplementary Fig. 4b). The addition of MΦ to the P1 culture increased the numbers of Emcn⁻ cells while just preserving the numbers of the Emcn⁺ population (Fig. 5e, f). Next, we tested whether the addition of M-CSF impacted BMEC outputs. As shown in Fig. 5g, M-CSF resulted in a ~5-fold increase of total BMEC (average 4-fold and 10-fold increase in Emcn⁺ and Emcn⁻, respectively).

Collectively, these results indicate that MΦ promote BMEC expansion and maintenance of their heterogeneity in ex vivo culture, whereas MCs exert an inhibitory effect.

## Emcn⁺ BMECs contribute to angiogenesis and support HSPCs

Next, we assessed the ability of both Emcn⁺ and Emcn⁻ BMECs to engage in in vitro angiogenesis in Matrigel tube assay. Emcn⁺ BMECs (sorted at either P0 or P1) were not able to form any type of structure in Matrigel (Fig. 6a and Supplementary Movie 3). In contrast, Emcn⁻ BMECs sorted at P0 formed robust cord-networks on Matrigel by 36 h, which remained stable up to 47 h (Fig. 6a and Supplementary Movie 3). Interestingly, Emcn⁻ BMECs from P1, P2, and P3 exhibited a faster dynamic of cord-networks formation, by forming cords already at ~6 h and completing them between 12 and 24 h (Fig. 6b, Supplementary Fig. 5a, and Supplementary Movie 3). Cord-networks that were formed by Emcn⁻ cells at P1, P2, and P3 appeared less defined and were less complex across passages, as determined by measurement of their cellular connectivity (Fig. 6a–c, Supplementary Fig. 5a, and Supplementary Movie 3).

To date, it has not been possible to determine whether different types of primary BMECs contribute differently to in vivo angiogenesis or have distinct ability to support HSCs. To assess how Emcn⁺ and Emcn⁻ cells contribute to angiogenesis, we performed in vivo Matrigel plug assays. Either Emcn⁺ or Emcn⁻ cells were sorted from P0 WBM at day14, admixed with Matrigel and injected subcutaneously into the flank of syngeneic mice. Two months later, plug vascularization was evaluated by CD31 immunostaining and tdT fluorescence. Unlike Emcn⁻ cells, which were not able to form any blood vessels, Emcn⁺ cells successfully participated in forming new blood vessels with endogenous ECs (Fig. 6d). These results indicate that Emcn⁺, but not Emcn⁻ BMECs, contribute to angiogenesis in vivo.

A critical function of ECs in the BM is to support HSCs in the vascular stem cell niche[3,33]. This method provided the opportunity to compare the ability of Emcn⁺ and Emcn⁻ BMECs to sustain hematopoietic stem/progenitor cells (HSPC: lineage⁻Sca-1⁺cKit⁺, LSK) in vitro. Cocultures of sorted LSK cells over layers of sorted BMECs showed that Emcn⁺ but not Emcn⁻ BMEC, preserved LSK cells in the absence of stem cell factor (SCF). In the presence of SCF, Emcn⁺ BMECs maintained LSK cells at significantly higher numbers and promoted lower rates of myeloid differentiation (Fig. 6e, f). Thus, in our model, sinusoidal-like Emcn⁺ BMECs exhibit a better ability to support HSC.

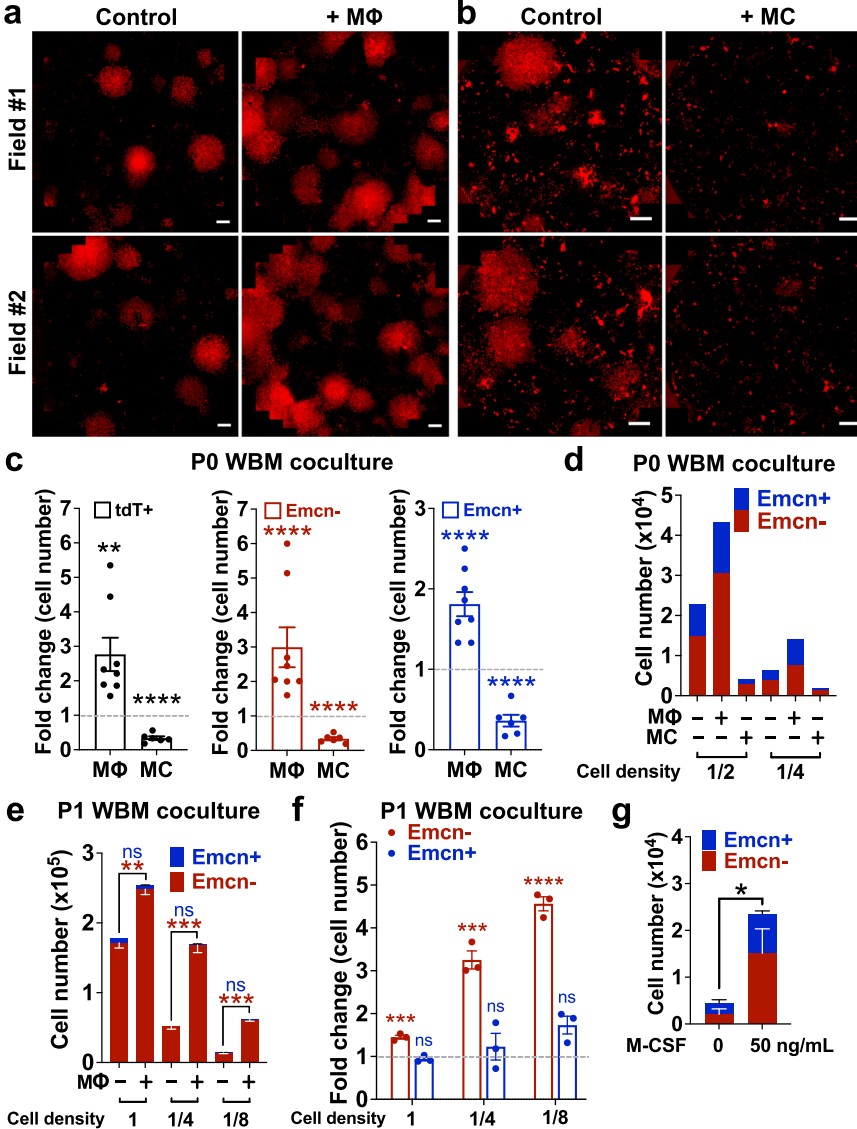

**Fig. 5 | Impact of HC/MΦ and MC on BMEC in WBM culture. a–d** P0 WBM cultures were divided in three conditions: control and supplement at day 0 with additional HC/MΦ or MC. **a, b** Snapshot images at day 14 culture showing tdT⁺ BMECs in P0 WBM cultures that were supplemented with **a** HC/MΦ or **b** MC, compared to control; two representative fields; Scale bar, 2000 μm. *n* = 12. **c** Fold change in BMEC numbers: tdT⁺ (left), Emcn⁻ (middle) and Emcn⁺ (right) when P0 WBM culture is supplemented with HC/Mϕ or MC, vs. culture control. Data are presented as the mean ± SEM. Emcn⁺ cells expanded on average 48-fold in control cultures (set up in this graph at value 1), 96-fold when supplemented with HC/MΦ, and 24-fold when supplemented with MC. BMEC + MΦ, *n* = 8; BMEC + MC, *n* = 6. *n* = 3 independent experiments. Statistics for all comparisons shown were determined using unpaired t-test, two-sided. tdT⁺ MΦ, *p* = 0.0026; tdT⁺ MC, *p* < 0.0001; Emcn⁻ MΦ, *p* < 0.0001; Emcn⁻ MC, *p* < 0.0001; Emcn⁺ MΦ, *p* < 0.0001; Emcn⁺ MC, ***p* < 0.001, and ****p* < 0.0001. Source data are provided as a Source Data file. **d** Total cell number of Emcn⁺ and Emcn⁻ cells of P0 WBM culture with or without additional HC/MΦ or MC at different WBM cell densities. Representative of *n* = 3

independent experiments. Source data are provided as a Source Data file. **e, f** P1 WBM cultures were supplemented with additional HC/MΦ and cultured for seven days: **e** total cell number and **f** fold changes of Emcn⁺ and Emcn⁻ cells supplemented with additional HC/MΦ vs. control, at different cell densities; *n* = 3. Data are presented as the mean ± SEM. Statistics for all comparisons shown were determined using unpaired t-test, two-sided. **e** Cell density 1, Emcn⁻, *p* = 0.0020; cell density 1/4, Emcn⁻, *p* = 0.0006; cell density 1/8, Emcn⁻, *p* = 0.0001; Emcn⁺ *p* = ns at all densities. **f** Cell density 1, Emcn⁻, *p* = 0.0006; cell density 1/4, Emcn⁻, *p* = 0.0004; cell density 1/8, Emcn⁻, *p* < 0.0001; Emcn⁺ *p* = ns at all cell densities. ***p* < 0.01, ****p* < 0.001, and *****p* < 0.0001. Source data are provided as a Source Data file. **g** P0 WBM cultures were supplemented with recombinant M-CSF (50 ng/mL) and cultured for fifteen days. Total cell number of Emcn⁺ and Emcn⁻ cells of P0 WBM culture with or without M-CSF; *n* = 3 independent cultures. Data are presented as the mean ± SEM. Statistics were determined using unpaired t-test, two-sided **p* = 0.0336.

## Molecular signature of Emcn⁺ and Emcn⁻ BMECs ex vivo

We performed bulk RNA-seq analysis of Emcn⁺ and Emcn⁻ BMEC sorted from P0 WBM. Their transcriptome was compared to the ones from Emcn⁻ cells sorted from lung cultures and BM MSC (Figure 7a). P0 Emcn⁺ BMECs showed higher expression of conventional EC identity genes, *Cdh5* (VE-cadherin), *Pecam1* (CD31) and *Tek* (Tie2), and expressed the most well-known venous EC marker gene *Ephb4* and other venous/sinusoidal endothelial genes, such as *Stab2, Flt4, Epor,*

*Dnase1/3,* and *Angpt2* (Fig. 7b and Supplementary Fig. 6a). These genes were within the top 5% of variable genes across all samples. P0 Emcn⁻ BMECs expressed higher levels of known arterial endothelial genes, such *Serpinf1, Ace, Tmem100,* and *Ltbp4* and were enriched in genes expressed by mesenchymal cells such as *Gsn, Runx2,* and *Pdgfrb* (Fig. 7b and Supplementary Fig. 6b). Among the differentially expressed genes (Fig. 7a, Supplementary Fig. 6a, b, and Supplementary Table 5), we noted that P0 Emcn⁺ BMECs expressed markers associated

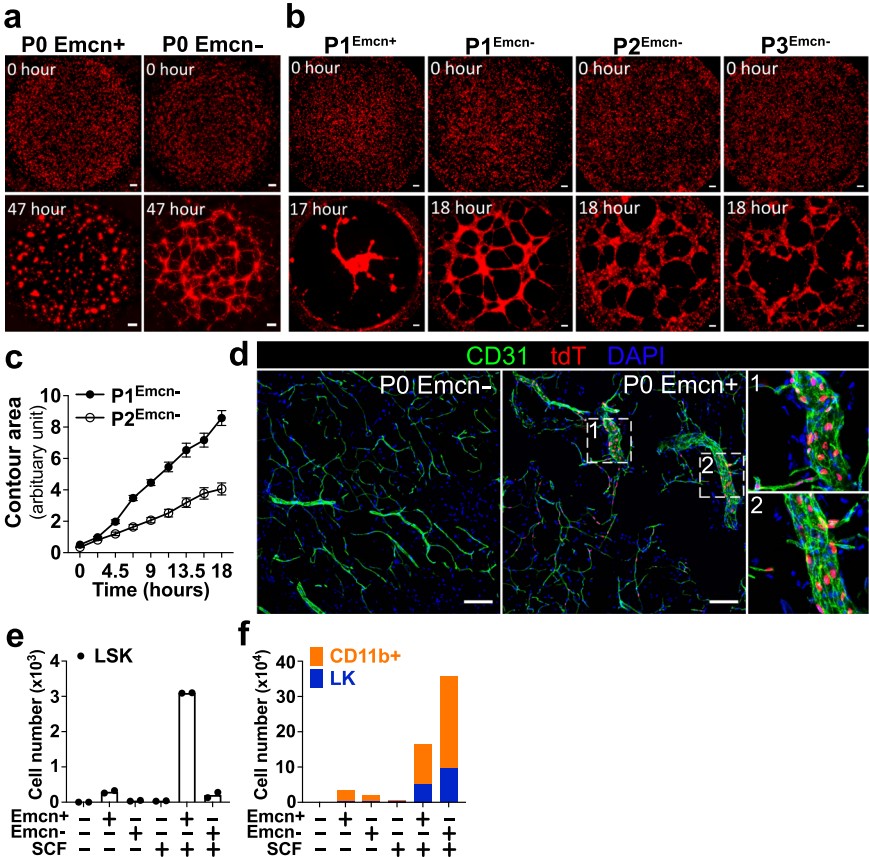

**Fig. 6 | Distinct functional properties of Emcn⁺ and Emcn⁻ BMEC. a, b** Time lapse images of cells in Matrigel at the indicated time points: **a** Emcn⁺ and Emcn⁻ cells sorted from P0 WBM cultures; **b** Emcn⁺ cells from P1$^{Emcn+}$ (grown in culture as sorted cells), and Emcn⁻ cells from P1$^{Emcn-}$, P2$^{Emcn-}$, and P3$^{Emcn-}$ (grown in culture as sorted cells). Scale bar, 200 μm. Representative of *n* = 8-12 independent experiments. **c** Quantification of cords formation and structure integrity by Emcn⁻ cells at P1 and P2. Line graph shows differences in growth patterns of cells at P1 and P2, indicated as a decrease in the contour area with passage (see details in Methods section). P1$^{Emcn-}$ *n* = 12; P2$^{Emcn-}$ *n* = 8. Data are presented as the mean ± SEM. Source data are provided as a Source Data file. **d** Immunostaining of CD31 marker on Matrigel plugs. Matrigel was admixed with sorted P0 Emcn⁻ (left) or P0 Emcn⁺ (right) cells and

implanted into the flanks of syngeneic mice for two months, prior collection and fixation. Note that only exogenous P0 Emcn⁺ cells (nucleus in red), but not P0 Emcn⁻ cells, can be observed in new blood vessels formed by endogenous ECs (nucleus not in red). Zoom-in image of white-dashed squares 1 and 2 are depicted on the right panel. Scale bar, 100 μm. Representative of *n* = 3 experiments. **e, f** Monolayers of Emcn⁺ or Emcn⁻ cells sorted from P0 WBM culture were cocultured with sorted LSK (6000 cells/well) for seven days in the presence or absence of SCF (50 ng/ml). Bar graphs show total cell numbers at day 7 for **e** LSK cells and **f** LK and CD11b⁺ cells; *n* = 2. Data are presented as the mean ± SEM. Source data are provided as a Source Data file.

with endothelial tip cells, such as *Apln, Cxcr4, Kdr, Dll4, Angpt2, Sox17,* and *Esm1* (Fig. 7c), whereas P0 Emcn⁻ BMECs expressed markers associated with endothelial stalk cells, such as *Eno2, Hk2, Meox2, Acss2, Aldh2,* and *Lgals3* (Fig. 7c and Supplementary Table 4). The molecular differences between P0 Emcn⁺ and P0 Emcn⁻ cells were even more evident when we restricted the analysis to signatures of 38 arterial/sinusoidal genes and 35 tip/stalk genes (Fig. 7b, c) that we compiled from the literature (Supplementary Table 6). Thus, P0 Emcn⁺ BMECs exhibited a combination of sinusoidal- and tip-like signature, whereas P0 Emcn⁻ expressed a combination of arterial- and stalk-like signature. These patterns of gene expression were also confirmed by RT-PCR (Fig. 7d). Of note, Emcn⁺ BMECs expressed higher levels of genes encoding for hematopoietic supportive molecules, such as *Dll4, Dll1, Tgfb1,* and *Pdgfb* (bottom panels of Supplementary Fig. 6a).

To sample differences between ECs from the BM and ECs from other organs, we analyzed lung ECs. We cultured mouse primary lung ECs following standard procedures[34] and found that they are predominantly constituted by Emcn⁻ cells. Comparison between BM- and lung-derived Emcn⁻ cells showed great similarity in their gene expression profile (Fig. 7a). BMECs and lung ECs did not express genes highly expressed in MSC (Supplementary Fig. 6c).

Analysis of Emcn⁻ BMECs transcriptomic profile during passages in vitro (Fig. 7e) shows that the transition from P0 to P1 was the one marked by most significant changes whereas the transitions from P1 to P4 exhibited a uniform gene expression profile. Major changes in the transition between P0 and P1 include a significant increase in the senescence marker p16 (*Cdkn2a*; Fig. 7g) and the downregulation of the Notch pathway (Fig. 7g and Supplementary Fig. 6d). Principal Component Analysis (PCA) of the expression matrix for all samples (including lung and BM MSC) was used to visualize relationships between populations. The plot of the first two principal components revealed that the transcriptomes of P0 Emcn⁺ BMECs, P0 Emcn⁻ BMECs, and BM MSCs, clustered in distinct areas; P0 Emcn⁻ BMECs and Lung Emcn⁻ ECs clustered together; P1-P4 Emcn⁻ BMECs transcriptomes clustered together and far from both P0 Emcn⁻ and P0 Emcn⁺ BMECs (Fig. 7f).

**BMEC heterogeneity and identity are preserved in culture**

An important question is whether BMECs in culture depart from their native, in vivo characteristics. To address this point, we compared the transcriptomes of fresh and cultured BMEC by single-cell RNA-seq (scRNA-seq) (scheme in Fig. 8a). First, we obtained single-cell transcriptomic data of 854 and 671 cells from two replicates, for a total of

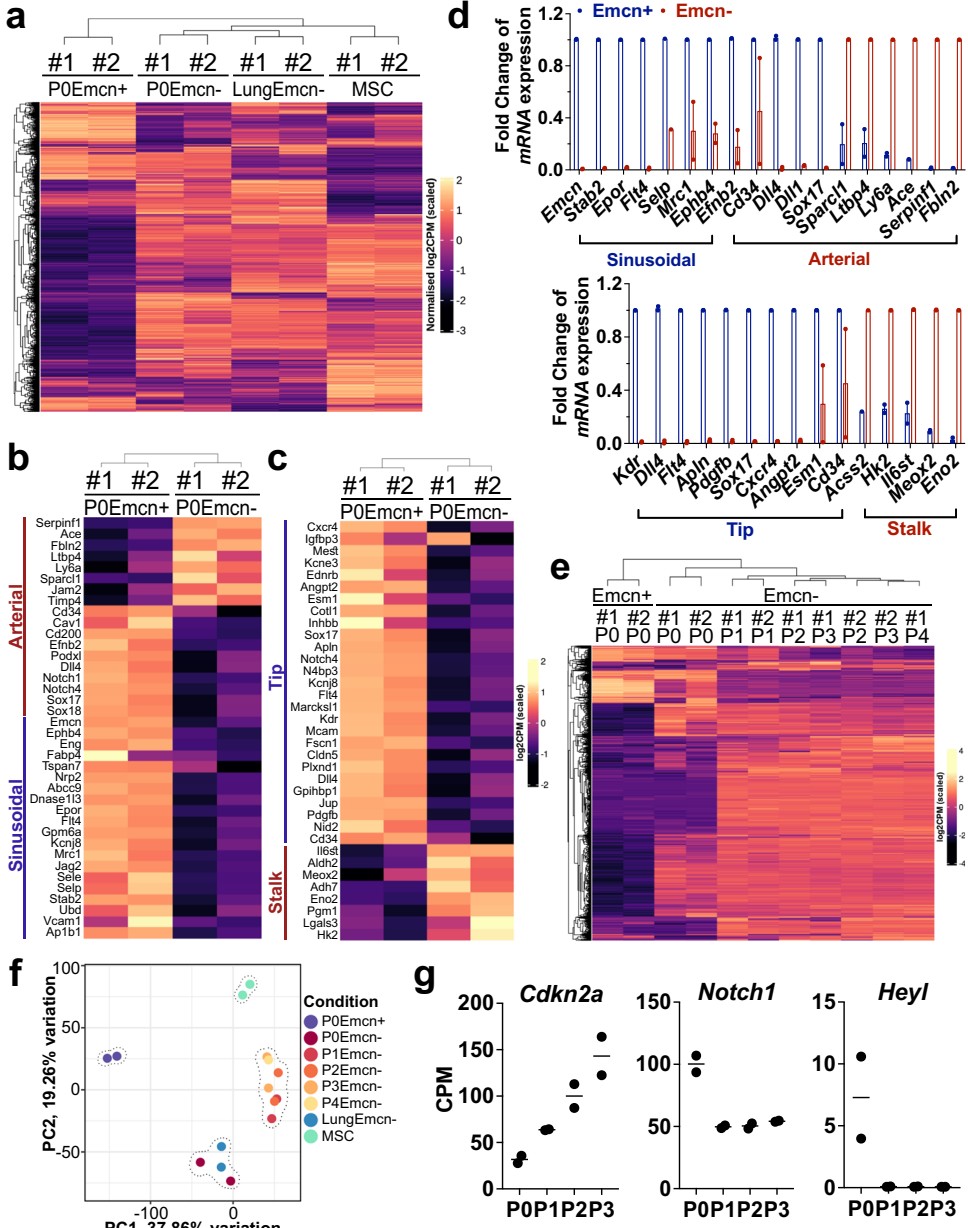

**Fig. 7 | Molecular Signatures of Emcn+ and Emcn− BMEC. a–c** Bulk-RNA-seq of sorted populations at day 14 of culture: Emcn+ and Emcn− BMEC from P0 WBM; Emcn− EC from lung; MSC from BM. **a** Heatmap of the top 5% variable genes. **b** Heatmap of arterial-sinusoidal gene signature and **c** tip-stalk signature. Measurements are expressed in scaled log2 counts per million (CPM). **d** Real-time (RT-) PCR for the indicated genes in Emcn+ and Emcn− BMEC from P0 WBM culture at day 1; *n* = 2. Source data are provided as a Source Data file. **e** Bulk-RNA-seq of Emcn− BMECs at the different passages. Heatmap of the top 5% variable genes. **f** Principal Component Analysis (PCA) of BMEC P0 Emcn+ and Emcn− and P1-P4 Emcn−; lung Emcn−, and MSC transcriptomes. **g** Analysis of the most differentially expressed genes in Emcn− BMEC cells during passages. Source data are provided as a Source Data file.

1525 tdT+ cells, BMEC, freshly isolated from BM. The endothelial identity of cells marked by Tie2CreERT2 was substantiated by integrating our data with publicly available data on populations marked by perivascular cell-specific LepR-Cre and osteoblast specific Col2.3-Cre[35] (Supplementary Fig. 7a, b). By clustering cells using PCA and a nearest neighbor approach (as in Seurat), we identified five clusters (clusters 0 to 4) (Fig. 8b). It is noteworthy that expression of *Emcn* clearly defined two populations within the fresh Tie2tdT+ cells: Emcn+ (clusters 1 and 3; *n* = 323 and 249) and Emcn− (cluster 0; *n* = 615) (Fig. 8b). Expression of *Tek* (Tie2) was evident in clusters 0, 1, and 3 but was below detection in cluster 2, which showed expression of the perivascular marker *Lepr* (*n* = 284 cells), and in cluster 4, which exhibited expression of the hematopoietic marker *Ptprc* (CD45) (*n* = 54 cells; Fig. 8c).

To define the cell types represented by each cluster, we visualized expression of gene signatures associated with arterial, sinusoidal, perivascular, and hematopoietic cells in a heatmap (Fig. 8d). Cluster 0 (Emcn−) highly expressed ~30% of the arterial markers represented in our compiled arterial/sinusoidal gene signature (Supplementary Table 5), such as *Serpinf1, Ace, Fbln2, Sparcl1, Ltbp4, Ly6a*, and *CD34* and did not express any of the sinusoidal associated genes (Fig. 8d). Thus, we are referring to cluster 0 as arterial-like. Cluster 1 (Emcn+) showed a 100% expression of the sinusoidal/venous markers presented in our signature, such as *Ephb4, Flt4, Stab2, Sele*, and others (Fig. 8d); we are referring to cluster 1 as a sinusoidal-like. Intriguingly, cluster 3 (also Emcn+) expressed ~30% of the sinusoidal markers and >80% of the arterial associated markers – exhibiting a

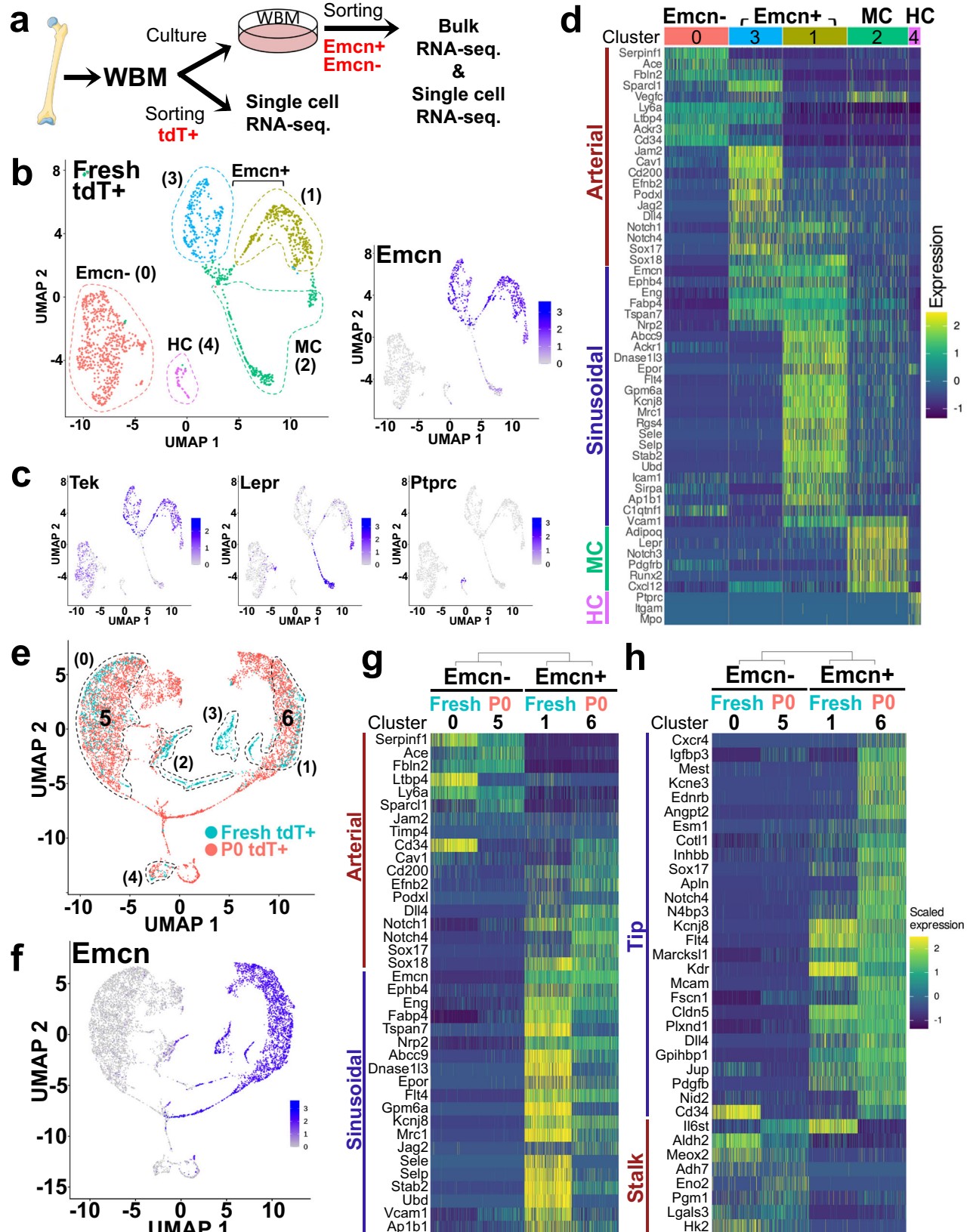

mixed but predominant arterial-like signature (Fig. 8d); we are referring to cluster 3 as a "mixed" cluster. Distribution of representative arterial (*Ly6a, Cd34, Sparcl1, Ltbp4, Podxl*) and sinusoidal (*Flt4, Stab2, Sele, Dnase1l3, Ephb4*) markers among clusters is shown in Supplementary Fig. 7c, d. Cluster 2 and 4 were tdT⁻ cells, likely contaminating populations from the sorting procedure and

exhibited signatures for perivascular cells and hematopoietic genes, respectively (Fig. 8d).

Next, we performed scRNA-seq of cultured tdT⁺ BMEC from P0 WBM culture (scheme in Fig. 8a). Following two weeks of WBM culture (P0), Emcn⁺ and Emcn⁻ were sorted and admixed together at 1:1 proportion, in duplicate. The mixed populations were subjected to scRNA-

**Fig. 8 | Native BMEC identity is preserved in ex vivo culture. a** Scheme of experimental design for freshly isolated BM tdT⁺ and P0 WBM cultured cells. Parts of the figure (the femur) were drawn by using pictures from Servier Medical Art by Servier, licensed under a Creative Commons Attribution 3.0 Unported License. **b**, **c** UMAP plot of single-cell gene expression profile from fresh cells. Clusters ranging from 0 to 4 identified by PCA and nearest neighbors are displayed in **b** using different colors. **c** Level of expression, reported as z-scaled log2 counts per million (CPM), of *Emcn*, *Tek*, *Lepr*, and *Ptprc* are shown in blue for each cluster in the UMAP visualization. **d** Gene expression profile for the sinusoidal/arterial gene signature. The levels of expression are reported as z-scaled log2 counts per million

(CPM). **e** Integrated UMAP plot of the single-cell gene expression profile of fresh sorted P0tdT⁺ (Emcn⁺ and Emcn⁻) (cyan) and cultured BMEC cells sorted at P0 (red). Numbers indicate clusters. **f** Level of *Emcn* expression, reported as z-scaled log2 counts per million (CPM), visualized in Integrated UMAP plot as in **e**. **g**, **h** Heatmaps of gene expression profiles in fresh and cultured BMECs using the arterial/sinusoidal signature (**g**) and the tip/stalk signature (**h**). Dendrograms above heatmaps show hierarchical clusters and the distance between the subpopulation's transcriptomes. Results show differences between Emcn⁻ and Emcn⁺ and the linkage between freshly isolated and cultured cells of the respective groups. The levels of expression are reported as z-scaled log2 counts per million (CPM).

seq, from which we recovered 6476 cells in total (2883 and 3593 cells from each replicate). The cultured P0ᵗᵈᵀ⁺ population (Emcn⁺ plus Emcn⁻) was visualized in an integrated UMAP analysis together with fresh BM tdT⁺ cells, now defining 7 clusters (Fig. 8e). Data showed clustering of cultured P0ᵗᵈᵀ⁺ cells with fresh BM tdT⁺ (cluster 5 and 0; cluster 6 and 1). UMAP visualization of Emcn expression (Fig. 8f), indicated that fresh and cultured Emcn⁻ BMECs clustered together and fresh and cultured Emcn⁺ BMECs clustered together. Examination of arterial/sinusoidal signature gene expression from fresh and cultured tdT⁺ BMEC cells showed remarkably similar profiles (Fig. 8g). We also determined the expression of genes defining the tip/stalk signature (Fig. 8h). This approach further -discriminated the Emcn⁺ from Emcn⁻ BMECs. To correct for possible bias in the choice of gene subset, we identified the top 5 percentile of the most variable genes (Supplementary Fig. 8a). The dendrogram structure above heatmaps (Fig. 8g, h, and Supplementary Fig. 8a) shows the linkage between fresh and cultivated cells of the respective groups (Emcn⁻ and Emcn⁺) and the separation between Emcn⁺ and Emcn⁻. Interestingly, we noted that a few genes flipped expression during the in vivo to in vitro transition (such as *CD34*, *Ltbp4*, and *Il6st*), and that Emcn⁺ BMEC acquired a stronger "tip" signature and expression of some "arterial" genes, likely due to the pro-angiogenic conditions of the in vitro culture (Fig. 8g, h). Surprisingly, there was no cultured BMEC cluster overlapping with fresh cluster 3, Emcn⁺ "mixed" (Fig. 8e), suggesting that this population is lost in culture.

Gene Set Enrichment Analysis (GSEA) in Emcn⁺ and Emcn⁻ BMEC revealed enrichment of pathways aligned with their distinct cell functions. Emcn⁺ BMEC are enriched in gene ontology (GO) pathways associated with cell migration, wound healing, sprouting and Rho proteins signals (Supplementary Fig. 8b). Such pathways well describe their "tip" behavior and the associated genes include "tip" markers such as *Cxcr4*, *Angp2*, *Apln*, *Flt4*, *Delta* ligands and *Notch* receptors (see Supplementary Table 5), *E2F* genes and *Cdc42*, *Rac*, *Rap*, and *Rho* family members. Emcn⁻ BMECs are enriched in pathways associated with external incapsulating structure organization, extracellular matrix dissembling, cell adhesion and growth factors (Supplementary Fig. 8b) that concord with their ability to form cords in Matrigel and their "stalk" behavior. Such pathways include most members of the Adam family of disintegrins and of the Mmp family of matrix metallopeptidase; hyaluronan synthases and collagens; Tgfβ pathway, and Fgf family members, and arterial identity signature genes *Efnb2, Sparcl, Caveolin*, and *il6st* (see Supplementary Table 6).

## Discussion

Lack of robust methods to define ECs' functions both in vivo and in vitro has limited the rigorous deployment of mouse models for mechanistic studies. Efforts to isolate and culture adult primary ECs from various organs by using immortalization strategies and genetic reprogramming[17,18] have been constrained by the challenge of expanding ECs without losing their heterogeneity and their native, primary, characteristics. Here, we used a *Tie2-CreERT2;Rosa26-tdTomato* reporter mouse to isolate, profile, and cultivate primary ECs from different organs using an ex vivo cell culture strategy that preserves some key cell-to-cell interactions and conditions of their

microenvironment in vivo and can harbor ECs with characteristics similar to primary cells in vivo.

In this study, we used BM as a paradigm. EC-specific Tie2 promoter-driven tdTomato expression was employed to track BMECs in a whole BM (WBM) cell culture system. This strategy allowed to demonstrate that primary BMECs can be cultured ex vivo without losing their heterogeneity and native molecular signatures. Notably, this approach enabled to culture sinusoidal and arterial BMECs and to map their functional properties, unveiling previously unrecognized features of these two populations (Summary in Fig. 9). WBM culture supports viability, proliferation, phenotype, and molecular profile of two BMEC populations that can be distinguished just by their distinct migratory behavior: one population forms 2D-networks and another forms cobblestone-like colonies. Expression of *Endomucin* (Emcn), a single transmembrane sialomucin expressed by capillaries and by venous but not arterial endothelium[22,23] is detected only on BMECs forming 2D-networks. Immunophenotypic and transcriptomic analysis indicated that Emcn⁺ BMECs exhibit a sinusoidal-like profile whereas Emcn⁻ BMECs exhibit an arterial-like profile.

We observed that Emcn⁺ BMECs show limited life span and proliferation activity, whereas Emcn⁻ BMECs exhibit high proliferative potential and expand beyond four passages in culture. It is intriguing that Emcn⁺ cells, which exhibit a high capacity for cell migration and organization in 2D, cannot engage in in vitro angiogenesis, whereas Emcn⁻ cells, which do not form networks in 2D, have instead a great ability to form cord-networks. Emcn overexpression has been shown to promote migration, proliferation, and cords formation in human retinal ECs[36]. Our discordant observations suggest that expression levels, tissue- and species-specificity are likely to influence the outcome of the studies. Interestingly, we observed that only Emcn⁺, but not Emcn⁻ BMECs, participate in neo-angiogenesis in vivo, by joining endogenous EC to form blood vessels. Collectively, this ex vivo culture system highlights some mutually exclusive properties of sinusoidal-like Emcn⁺ and arterial-like Emcn⁻ cells and indicates that the ability of BMECs to form 2D-networks and to engage in in vitro or in vivo angiogenesis are processes regulated by independent mechanisms.

A key feature of this culture system, associated with its ability to sustain BMECs diversity ex vivo, is the presence of BM macrophages. Collectively, our experiments demonstrate a critical impact of macrophages on BMECs expansion and on the preservation of Emcn⁺ BMECs. This positive impact of BM macrophages is not surprising, given their known role in vascular repair and regeneration[37]. Interestingly, it was unexpected that BM mesenchymal cells inhibited BMEC growth, given their positive role on HUVEC cultures[38]. Thus, our work shows the importance of macrophages in supporting BMEC longevity, most likely by secreted factors, and unveil a cell−cell mediated inhibitory effect of mesenchymal cells, providing a platform to address specific molecular mechanisms in the future.

In the BM, EC are a critical component of the hematopoietic stem cell (HSC) niche[6,33,39,40]. Attempts to recapitulate the interactions between BMECs and HSCs in vitro, particularly in mouse, have been challenged by the difficulty to isolate and maintain BMECs in vitro. This challenge has been partially circumvented by genetic manipulation approaches or by using EC from other organs[18,19]. To date, primary BM-

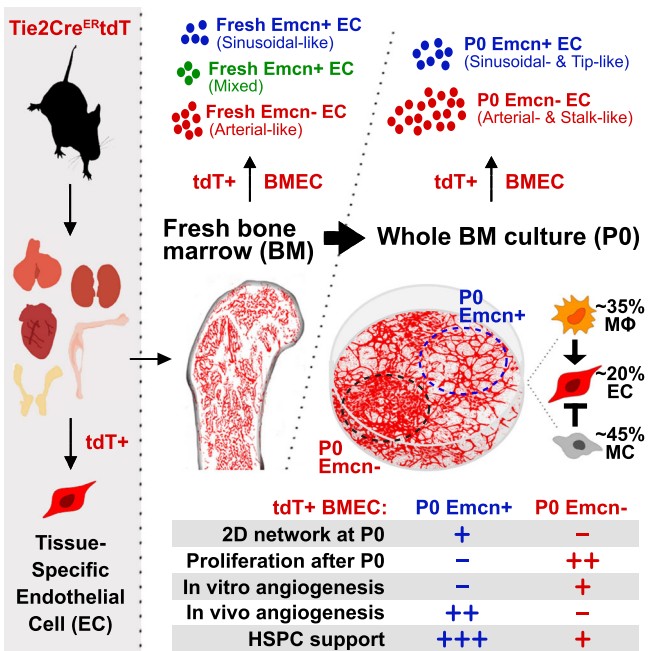

**Fig. 9 | Summary of WBM ex vivo culture workflow and BMEC distinct characteristics.** ECs can be isolated from all organs via their Tie2Cre-driven tdTomato expression. In vivo, three major BMEC populations can be identified in the BM by scRNA-seq transcriptomic profile: Emcn+ sinusoidal-like, Emcn- arterial-like and Emcn+ with mixed sinusoidal/arterial-like signature, which are no longer observed in ex vivo culture. BMECs require interactions with WBM to maintain their heterogeneity and their native molecular signatures ex vivo. Macrophages (MΦ) promote maintenance of Emcn+ sinusoidal-like BMEC and expansion of all BMECs, whereas mesenchymal cells (MC) have an inhibitory effect. Endomucin expression distinguishes BMEC in Emcn+ and Emcn- populations exhibiting distinct behaviors, as summarized in the table shown at the bottom of the figure.

derived arteriolar and sinusoidal BMECs have never been directly compared for ability to support hematopoiesis. Here, we compared side-by-side the ability of sinusoidal-like Emcn+ and arterial-like Emcn- BMECs to support HSPCs by using coculture experiments. We found that Emcn+ ECs proved to be superior in supporting maintenance of immunophenotypic HSPC in vitro. Not surprising, this property was associated with higher expression of Notch ligands, known to promote HSC maintenance[41,42].

A critical question in our studies was whether BMECs cultured ex vivo drift from their fresh counterpart losing their native identity. scRNA-seq analysis identified 3 major clusters in BMECs in vivo: (1) Emcn- arterial-like ECs, (2) Emcn+ sinusoidal-like ECs, and (3) Emcn+ "mixed" ECs, which expressed both arterial and sinusoidal genes. Hierarchical clustering analysis shows the linkage between fresh and cultivated cells of the respective groups (Emcn+ and Emcn-), indicating a significant similarity in their in vivo and ex vivo transcriptome. Sinusoidal and arterial signatures are remarkably well conserved in Emcn+ and Emcn- during ex vivo culture. Application of a "tip/stalk" signature, which characterizes the function of sprouting during vascular morphogenesis[43], further defines Emcn+ cells as "tip-like" cells and Emcn- cells as "stalk-like" cells, in both fresh and cultured cells. The observation of a "tip-stalk" signature in fresh cells from homeostatic vessel beds, which have no or little sprouting behavior, is intriguing. One speculation is that adult vessels at homeostasis maintain a baseline expression of a restricted group of "tip/stalk" genes – perhaps memory of a sprouting occurred – that can be poised to be induced to higher expression when a situation necessitating regeneration occurs. Despite their global transcriptomic similarity, in vivo and ex vivo BMEC are not identical. We observed that during ex vivo culture, the expression of sinusoidal and arterial markers became attenuated

whereas the expression of "tip" and "stalk" markers became accentuated in Emcn+ and Emcn- cells, respectively. This data suggests that a "tip" and "stalk" polarization of Emcn+ and Emcn- BMEC occurs in culture, likely due to the angiogenic conditions of our ex vivo culture system that may artificially mimic a regenerative call.

Despite the lower representation of arterial-like Emcn- ECs in fresh BM compared to sinusoidal-like Emcn+ ECs, and their limited proliferation in vivo, Emcn- cells exhibit high proliferative potential in vitro, similarly to endothelial cell colony-forming units (ECFCs)[20,21,44,45]. It is possible that these cells, relative quiescent at steady state, can be activated during process of vessel regeneration in response to disease or injury and that the presence of angiogenic factors in culture triggers a similar proliferative/regenerative response in vitro. Interestingly, Emcn- BMECs share similarity with the transcriptomic profile of an ECs population harvested from the melanoma tumor microenvironment, recently described by Khosrotehrani's group[46,47] and named as Endothelial Vascular Progenitors (EVP). This elegant study[46] showed that these EVP can give rise to more differentiated endothelial cells and contribute to neo-angiogenesis in vivo. However, in our study, we did not find evidence of Emcn- generating Emcn+ ECs or contributing to in vivo angiogenesis. How the in vitro attributes uncovered here for Emcn- BMECs relate to their in vivo properties remains elusive. We anticipate that our platform will help to elucidate the nature of these cells in future studies.

Here, we have observed that expression of Endomucin discriminates distinctively, and more accurately than Sca-1 expression, BMEC with sinusoidal- or arterial-associated genes. However, our approach has also unveiled another type of Emcn+ population, which expresses both arterial and sinusoidal genes. This rare population is observed only in vivo but it is no longer observed in ex vivo culture. This Emcn+ "mixed" population, identified by scRNA-seq, exhibits a dual arterial/sinusoidal transcriptomic signature and may represent a transitional state from a common precursor or a potential ancestor of Emcn+ and Emcn- population(s). Studies assessing the potential and functional characteristics of this subset, are warranted.

In conclusion, we have developed a robust ex vivo method allowing to culture and compare two major EC subtypes in the BM: arterial-like and sinusoidal-like endothelial cells. Prospectively, this approach provides a valuable and amenable platform to perform mechanistic studies of processes like sprouting, angiogenesis, cell plasticity, and cellular crosstalk, which can be paired with drug screening and in vivo studies. Ultimately, this model represents a simple and effective tool enabling the use of primary ECs to study tumor angiogenesis and vascular regeneration in different pathologies and conditions.

## Methods
### Animals
Animal experiments were performed in compliance with the ethical regulations of the City of Hope Institutional Animal Care and Use Committee (IACUC).

Experiments were conducted in 4-6 months-old Tie2-CreER-T2;Rosa26-tdTomato and Osx-Cre;Rosa26-tdTomato mice under the City of Hope IACUC 20023 approved protocol. We did not observe sex-associated differences in the isolation and characteristics of BM-derived EC. Thus, animals were used regardless of their sex.

### Mouse generation
This specific *Tie2-CreERT2* mouse strain was developed by Dr. Yi Zheng's laboratory. It has been generated by using the DNA fragment containing intron sequence of *β-globulin* gene, Cre recombinase, mutated ligand binding domain of the estrogen receptor and SV40 polyA (klenow enzyme filled Stu I site- Xba I DNA fragment) excised from plasmid pCreER^T2 and inserted into the EcoRV-Xba I site of plasmid pBSTie2-2. The resulting plasmid was called pTie2P-CreER^T2.

The Sal I fragment containing *Tie2* promoter and CreERT2 was excised from pTie2P-CreER$^{T2}$ and cloned upstream into the SV40 polyA signal and the first intron sequence of pBSpolyATie2-10 to generate the transgenic construct. The final *Tie2-CreER$^{T2}$* transgene, comprising the *Tie2* promoter, *β-globulin* intron sequence, *CreER$^{T2}$* fusion gene, 2 polyadenylation signal (pA) fragments, and *Tie2* enhancer sequence was excised from the vector and used for C57BL/6 pronuclear micro-injection to generate the transgenic mouse line.

*Tie2-CreERT2;Rosa26-tdTomato* mice were generated by cross-breeding *Tie2-CreERT2* mice with *Rosa26-tdTomato* mice (B6;129S6-*Gt(ROSA)26Sor$^{tm9(CAG-tdTomato)Hze}$*/J); *Ai9* or *Ai9* (*RCL-tdT*); Jackson Laboratory stock number: 007905). 25 μg/g of Tamoxifen (Sigma, T5648) were injected into 2–7-week-old *Tie2-CreERT2;Rosa26-tdTomato* mice once a day for 5 days.

*Osx-Cre;Rosa26-tdTomato* mice were used to obtain bona-fide *Osx-Cre*-mediated tdT$^+$ MSC for bulk RNA-seq. *Osx-Cre* mice[48] were obtained from Dr. Ernestina Schipani. *Osx-Cre;Rosa26-tdTomato* mice were generated by crossbreeding *Osx-Cre* mice with *Rosa26-tdTomato* mice (as above, Jackson Laboratory stock number: 007905).

### Primary EC isolation and culture from tissues

Lungs, kidneys, hearts, spleens, livers, or SAT (subcutaneous adipose tissues) were harvested and washed with a copious amount of sterile ice-cold 1X PBS to remove the maximum amount of blood cells from the tissues. The washed tissues were minced using a scalpel blade or micro-dissection scissors in 60 mm sterile Petri plates under a sterile hood, incubated with a digestion buffer containing DMEM supplemented with 20 mM HEPES, 3 mg/mL collagenase I, BSA 1 mg/mL, and 0.1 mg/mL DNase I for 30 min at 37 °C with gentle agitation (0,24 g). The digested tissues were filtered via a 70 μm cell strainer. The supernatant was collected and centrifuged at 135 g for 5 min. Cell pellets were dissolved in Endothelial Cell Growth Medium-2 (EGM-2) BulletKit (Lonza, CC-3156 & CC-4176) including EBM-2 Basal medium, FBS 10 mL, Hydrocortisone 0.2 mL, hFGF-B 2 mL, VEGF 0.5 mL, R3-IGF-1 0.5 mL, Ascorbic Acid 0.5 mL, hEGF 0.5 mL, GA-1000 0.5 mL, Heparin 0.5 mL, and 1% Penicillin/Streptomycin. Cell suspensions were plated on collagen type I-coated (Corning) or 0.1% gelatin (Millipore)-coated culture dishes and cultured at 37 °C with 5% CO2 in an incubator (Panasonic). One mouse usually generates one 10 cm dish for lung, kidney, and heart, and one 6 cm dish for SAT. After 2 days, half (v/v) medium was added. On the day 4–5, half of cell suspension (v/v) was removed from the culture and replenished by fresh EGM-2 medium. Half (v/v) medium changes were performed every 2–3 days for approximately 2 weeks until confluent.

### Primary BMEC culture

Protocol to harvest fresh *tdTomato$^+$* ECs from mouse BM: Femurs and tibias isolated from *Tie2-CreERT2;Rosa26-tdTomato* mice were collected into sterile cold PBS with 2% FBS and gently crushed using a mortar and pestle. To avoid damaging the ECs, especially those ones localized just beneath the bones, crushing was performed briefly and very gently after reducing each bone into 2-3 pieces. Note that different strengths in crushing can lead to different results. Cell suspensions were then filtered into a 50 mL tube on ice using a 70 μm cell strainer and resuspended in 5 mL of EGM-2. The remaining crushed bone pieces were collected into a 15 mL tube and digested for 10 min in 1–2 mL/mouse bone digestion buffer (containing DMEM supplemented with 20 mM HEPES, 3 mg/mL collagenase I, BSA 1 mg/mL, and 0.1 mg/mL DNase I) at 37 °C and with gentle agitation (0,24 g). The rack in the shaking incubator was adjusted in a such way that the 15 mL tubes containing the crushed bones were positioned at a 45 degrees angle. After 10 min, the bone digested mixture was brought to a 10 mL volume in PBS containing 2% FBS, filtered with a 70 μm cell strainer and added to the cell suspension previously prepared into the 50 mL tube. A second digestion was repeated on the bone pieces in 1–2 mL of

digestion media for 30 min at 37 °C with stronger agitation (0,42 g) and at a 45 degrees angle. The cells suspension derived from the bone digestion was washed again in PBS with 2% FBS and added to the previous 50 mL tube after filtration through the same 70 μm cell strainer on ice. The digested bone pieces were collected back into the mortar and further crushed in PBS with 2% FBS. This crushing was performed more vigorously to harvest all the hematopoietic cells from the bones. The supernatant was passed through the same 70 μm cell strainer and pooled into the same 50 mL tube, on ice. PBS with 2% FBS was added through the same cell strainer to fill the 50 mL tube up to 50 mL. The whole cell suspension was centrifuged at 135 g for 5 min and the cell pellet was resuspended in EGM-2 medium for counting.

On average, the number of BMECs (CD45$^-$Ter119$^-$tdT$^+$ cells) was $1 \times 10^4$ cells/mouse (range $8 \times 10^3$–$11 \times 10^3$; Supplementary Table 1 and 2). On average, the number of BMEC sorted per mouse was 5598 (Supplementary Table 3).

WBM Culture: All cells obtained and pooled from crashed and digested bones (WBM cells), as described above, were resuspended in EGM-2 media, plated on collagen type I-coated (Corning), or 0.1% gelatin, (Millipore)-coated culture dishes at the density of $6$–$8 \times 10^7$/10 cm dish, and cultured at 37 °C with 5% CO$_2$ in an incubator (Panasonic). Usually one mouse (2 femurs and tibias) was necessary for one 10 cm dish. After 2 days, half (v/v) medium was added. On the day 4–5, half of cell suspension (v/v) was removed from the culture and replenished by fresh EGM-2 medium. Half (v/v) medium changes were performed every 2-3 days for approximately 2 weeks until confluent. At that point, the culture, consisting entirely of adherent cells, was gently trypsinized using 0.05% Trypsin/0.53 mM EDTA (Corning) and replated at a density of $1 \times 10^6$/10 cm dish. Cultures were passaged every 1–2 weeks at confluency. All passages and sorted cells from P0 were cultured for 1–2 weeks until confluent with half medium changes every 2–3 days. Note that results may vary if a different endothelium medium is used.

Cultures of sorted BM tdTomato+ BMECs: tdTomato$^+$ BMECs (T0$^{tdT+}$) sorted from fresh BM were directly seeded at the density of ~5000 cells/well in 24-well-plates and cultured in EGM-2 medium at 37 °C with 5% CO$_2$ in an incubator (Panasonic) by changing half (v/v) medium every 2–3 days for around 2 weeks until confluent.

Addition of supplementary HC/MΦ or MC into P0 WBM culture: CD11b$^+$ cells were isolated from fresh BM using EasySep™ Mouse FITC Positive Selection Kit II (STEMCELL). In each well of 12-well-plates ~$2 \times 10^6$ or ~$1 \times 10^6$ WBM cells were supplemented with $4 \times 10^6$ CD11b$^+$ cells (equivalent to a final density 1/2 and 1/4, respectively). DAPI$^-$CD45$^-$tdTomato$^-$ cells (MC) were sorted from a P0 culture at day 14 and ~$1.5 \times 10^5$ were added to ~$2 \times 10^6$ or ~$1 \times 10^6$ WBM cells (equivalent to density 1/2 and 1/4, respectively). Cells in suspension were removed at day 2 and 4 of culture, in such a way that only adherent CD11b$^+$ cells were selected. Cultured cells were detached at day 14 for cell counting and FACS analysis. Following CD11b$^+$ cell supplement, the number of MΦ in the WBM at day 14 was increased ~2-fold compared to standard WBC cultures, while the number of MC was enhanced ~4-fold. Wells at low density did not reach confluency. In experiments using a transwell, the same numbers of the isolated CD11b$^+$ cells or MC were added to the transwell membranes posed over 12 well/plate wells seeded with WBM (as above) and cultured until day 14.

Addition of supplementary HC/MΦ into P1 WBM culture: CD11b$^+$ cells were isolated as described above and plated at a density of $1 \times 10^7$/well in 6-well-plates. Cells in suspension were removed on the next day while the adherent CD11b$^+$ cells were combined with $1.8 \times 10^5$, $4.5 \times 10^4$, or $2.25 \times 10^4$ adherent cells harvested from WBM P0 at day 14 (cell density 1, 1/4, and 1/8, respectively, as indicated in Fig. 5e, f) and cultured for 2 weeks (P1).

HSPC coculture: DAPI$^-$CD45$^-$tdTomato$^+$Emcn$^+$ and DAPI$^-$CD45$^-$tdTomato$^+$Emcn$^-$ cells sorted from day 14 P0 culture were seeded at a density of $5 \times 10^4$/well in 12-well-plates for approximately

3 days (until confluence). LSK cells (~6000 cells) sorted from fresh BM of adult wild-type mice (2–5 months old) were added to the culture for 7 or 14 days in the absence or presence of SCF (50 ng/ml).

## Primary BM mesenchymal cell culture

Femurs and tibias isolated from *Osx-Cre;Rosa26-tdTomato* mice were collected into sterile cold PBS with 2% FBS and gently crushed using a mortar and pestle. Cell suspensions were then filtered into a 50 mL tube on ice using a 70 μm cell strainer while the crushed bone pieces were incubated at 37 °C for 40 min with gentle agitation (0,24 g) in 1–2 mL/mouse of bone digestion buffer containing DMEM supplemented with 20 mM HEPES, 3 mg/mL collagenase I, BSA 1 mg/mL, and 0.1 mg/mL DNase I. The cells suspension derived from the bone digestion was washed again in PBS with 2% FBS and added to the previous 50 mL tube after filtration through the same 70 μm cell strainer on ice. The digested bone pieces were collected back into the mortar and further crushed in PBS with 2% FBS. The supernatant was passed through the same 70 μm cell strainer and gathered into the same 50 mL tube on ice. The whole cell suspension was centrifuged at 135 g for 5 min and the cell pellet was dissolved in Mesencult medium (MesenCult™ Expansion Kit -Mouse; Stem Cell Technologies; #05513) and plated on collagen type I-coated (Corning) or 0.1% gelatin (Millipore)-coated culture dishes at the density of $6$–$8 \times 10^7$/10 cm dish (usually 2 legs per mouse per 10 cm dish) and started to be cultured at 37 °C with 5% $CO_2$ in an incubator (Panasonic). After 2 days, half (v/v) medium was added. On the day 4–5, half of cell suspension (v/v) was removed from the culture and replenished by fresh Mesencult medium. Half (v/v) medium changes were performed every 2-3 days for ~2 weeks until confluent. At that point, the culture, consisting entirely of adherent cells, was gently trypsinized using 0.05% Trypsin/0.53 mM EDTA (Corning) and DAPI⁻CD45⁻tdTomato⁺ cells were sorted by FACSAria Fusion (BD Biosciences).

## Flow cytometry and cell sorting

Fresh BMEC cells were harvested as described above. At day 14, P0 cultured cells were detached by Trypsin-EDTA (0.05% trypsin, 0.02% EDTA) for 5 min and further detached by pipetting. Cells were then suspended in PBS containing 2% FBS at $1 \times 10^7$ cells/mL and stained for 30 min on ice with the following antibodies: APC-anti-CD45 plus APC-anti-Ter119 (to eliminate hematopoietic and erythroid cells), and Alexa-488-anti-Emcn. DAPI⁻CD45⁻Ter119⁻tdTomato⁺ cells from fresh BM and the following populations from P0 cultured cells: DAPI⁻CD45⁻tdTomato⁺Emcn⁺, DAPI⁻CD45⁻tdTomato⁺Emcn⁻, DAPI⁻CD45⁻tdTomato⁻, and DAPI⁻CD45⁺tdTomato⁻ were sorted by FACSAria Fusion (BD Biosciences). Cell number yields of sorted fresh or cultured BMEC populations are summarized in Supplementary Tables 7 and 8. Complete characterization of BMECs was performed by using the following antibodies: CD45, Ter119, CD31, CD144, CD140a, CD105, Sca-1, Emcn, Flk-1, CD14, F4/80, and CD11b. Cells were acquired by LSRII (BD Biosciences) and analyzed with FlowJo. For LSK sorting, DAPI⁻Lin⁻Sca-1⁺cKit⁺ cells from lineage-depleted fresh BM cells using EasySep™ Mouse Hematopoietic Progenitor Cell Isolation Kit (STEMCELL) were sorted by FACSAria Fusion (BD Biosciences). 1 μL of each antibody was used per $1 \times 10^7$ cells. Detailed information of antibodies is shown in Supplementary Table 9. Gating strategies are shown in Supplementary Fig. 9.

## Immunofluorescence staining

*Frozen sections*. Bones were isolated and immediately fixed overnight in cold 4% paraformaldehyde/PBS at 4 °C. The following day, bones were washed 3 times for 5 min with cold PBS and decalcified in cold 0.5 M EDTA (pH 7.4–7.6) at 4 °C for 24-48 h with gentle agitation. Subsequently, the decalcified bones were washed three times with cold PBS and cryoprotected in PBS containing 20% sucrose and 2% PVP at 4 °C

for 24-48 h. Bones were then embedded in PBS containing 8% gelatin, 20% sucrose, and 2% PVP, incubated at RT for 30 min, and transferred to −80 °C for storage. Frozen bones were cut into sections with a thickness of 70–100 μm using a Leica CM3050-S cryostat and high profile cryotome blades. Frozen sections were dried at RT for 30 min, rehydrated with PBS at RT for 5 min, permeabilized by PBS containing 0.3% (v/v) Triton X-100 at RT for 20 min, incubated in the blocking solution made of PBS containing 10% goat or donkey serum, 5% BSA and 0.01% Triton X-100, at RT for 1 h. The slides were incubated with primary antibodies at 4 °C overnight, washed with PBS, and incubated with secondary antibodies at RT for 75 min. DAPI staining was performed at RT for 30 min after PBS washing. VECTASHIELD Antifade Mounting Medium (Vector Laboratories) was used for mounting. Images were captured by a confocal microscope LSM880 (Zeiss) and analyzed by a ZEN (Zeiss) software. For immunostaining, all primary antibodies for Emcn, CD31, Sma, VE-Cadherin, and Laminin were used at 1:100 dilution, and all secondary antibodies at 1:400 dilution. Detailed information of antibodies is shown in Supplementary Table 9.

## Time lapse microscopy and image analysis

A 3.5 cm culture dish for monitoring cell growth of P0 WBM culture was placed in the chamber of an inverted Axio Observer 7 microscope (Carl Zeiss) at 37 °C and 5% $CO_2$. Tiled images with z-slices were acquired as 20 per tile every 45 min for 8 days (day 5–13 of P0 WBM culture). The acquired images were stitched and processes with orthogonal projections for maximum intensity using ZEN (Zeiss) software and included as Supplementary Movie 1–3.

## In vitro Matrigel tube (cord) formation assay and analysis

Each well of μ-Slide Angiogenesis (Ibidi) was coated with 10 μL of Growth factor-reduced Matrigel (Corning) on ice and placed in the humidified chamber at 37 °C for 1 h to be polymerized. Cells were counted, suspended in the EGM-2 medium at a concentration of $3 \times 10^5$/mL, and carefully applied 50 μL into the Matrigel-coated well. The Lid-covered μ-Slide Angiogenesis was incubated at 37 °C and 5% $CO_2$ for 24–48 h. Tiled images with z-slices were acquired as 20 per tile every 45 min, stitched, and processed with orthogonal projections for maximum intensity using ZEN (Zeiss) software and made as Supplementary Movie 3.

Original images obtained by time lapse microscopy for all observation times underwent contrast-limited adaptive histogram equalization, cropping of the growth boundary, and gaussian blurring. Next, images were converted to binary and skeletonized[49] through a process of repeated erosion operations until prior to disconnection. This allowed kinematic analysis of endothelial cell cords/tube formation by quantification of changes in cellular connectivity. The network metric analyzed for quantification is the distribution of areas enclosed by fully connected network cycles (i.e., areas contained in loops of connectivity). Alterations in connectivity, due to either less formation of complete enclosures in the matrix, or thickening in the matrix connections), result in a smaller measure of enclosed area and expressed as arbitrary units. This metric is accessible using the pre-packaged contour detection functionality of the open-source software OpenCV.

## In vivo Matrigel plug assay

$2 \times 10^5$ BM ECs were mixed with 400–500 μL of Standard Matrigel (Corning) on ice. The EC-containing Matrigel was slowly injected subcutaneously into the right-shaved flank of the mouse, which was anesthetized with 5% of inhalant isoflurane and kept under 3% isoflurane anesthesia during all procedure. The left side flank of the same mouse was used for Matrigel injection without any cell as internal control. After 2 months, the Matrigel plugs were collected, trimmed by removing connective and adipose tissues surrounding the plugs, briefly washed by PBS, subjected to imaging, embedded into OCT compound, and stored at −80 °C for the preparation of frozen section.

### RNA extraction and qRT-PCR

Total RNAs were extracted using Trizol (Invitrogen) according to the manufacturer's instructions. The RNAs were converted into cDNA using the iScript cDNA Synthesis Kit (Bio Rad). qRT-PCR was performed with FastStart Essential DNA Green Master (Roche) using Light Cycler 96 (Roche). Primer information is shown in Supplementary Table 10.

### Bulk RNA-seq and scRNA-seq

RNA-seq and scRNA-seq libraries were prepared as previously described[50]. For RNA-seq, total RNAs were extracted using the Trizol reagent. RNA integrity and quantity were determined by using the RNA screentape on a 4200 TapeStation system (Agilent). Samples with RNA integrity number above 7 were used for library construction. The RNA-seq libraries were generated with the KAPA RNA HyperPrep Kit (Roche, no. KK8581). Libraries were sequenced on an Illumina HiSeq X10 (for the 1$^{st}$ batch of biological replica) and Illumina Hiseq2500 with Rapid mode (for the 2$^{nd}$ and 3$^{th}$ batch of biological replica). For scRNA-seq, DAPI$^-$CD45$^-$Ter119$^-$tdTomato$^+$ sorted cells from the fresh (without culture) bone marrows of two 4-month-old *Tie2-CreERT2;Rosa26-tdTomato* mice were loaded on the Chromium controller (10XGenomics) based on the number provided by the sorting equipment. Libraries were prepared using the Chromium Single Cell 3′ Reagent Kits (v3): Single Cell 3′ GEM, Library & Gel Bead Kit v3 (PN-1000075), Chromium Chip B Single Cell Kit (PN-1000073) and i7 Multiplex Kit (PN-120262) (10x Genomics) and following the Chromium Single Cell 3′ Reagent Kits (v3) User Guide (manual part no. CG000183 Rev A). Libraries were run on an Illumina HiSeq 2500 as 150-bp paired-end reads, at 20,000 read pairs per cell.

Bulk RNAseq analysis: To assess the quality of the sequencing libraries, the default parameters of the nf-core RNASeq pipeline, v1.4.2[51] were used. Briefly, trimmed reads were mapped to the GRCm38 reference using *STAR*[52] and abundance of transcripts from the Ensembl annotation (v81) was estimated using Salmon[53]. Each library was subjected to extensive quality control, including estimation of library complexity, sequence quality, read length and depth, among other metrics detailed in the pipeline repository (https://nf-co.re/rnaseq). Pseudocounts were merged into a matrix of counts per million per gene for each sample using the tximport R package[54]. Heatmaps were drawn using the ComplexHeatmap package[55] after transforming counts with the vst function from DESeq2[56]; the same transformation was used to visualize samples in PCA space. Differential expression was assessed using the default parameters of DESeq2. We called differentially expressed genes using merged counts with a Benjamini-Hochberg false discovery rate cutoff of 0.05 and absolute shrunken log2-fold change (LFC) > 1.5 between P0 Emcn$^-$ and each of the remaining samples. Shrunken LFC was obtained using the ashr adaptive shrinkage estimator implemented in DESeq2[57].

Single-cell RNAseq analysis: Processing of the scRNAseq reads was performed using the 10XGenomics Cell Ranger pipeline (v5.0.0) and visualized in R with the Seurat package (v4.0.5)[58]. Briefly, FASTQ files were generated using cellranger mkfastq with default parameters and cellranger count was run using the reference genome and annotation supplied (mm10-2.1.0) for each sample, counting UMIs for each gene. The output for each sample was integrated using cellranger aggr with default options and imported into R using Seurat for further analysis. Data for perivascular (*Lepr*-expressing) and osteoblast (*Col1a1*-expressing) cells were downloaded from the Gene Expression Omnibus (GSE108891) and imported to R. We kept cells where mitochondrial transcripts were <20%, > 200 genes expressed, <5% ribosomal transcripts, and genes expressed > 3 cells. To control for batch effects across experiments and between laboratories, the top 2000 variable genes shared by all samples under consideration were identified using the SelectIntegrationFeatures function. Integration anchors were identified based on these genes using the FindIntegrationAnchors

function, employing the log normalization method. Samples were then integrated using the IntegrateData function. Principal component analysis (PCA) and uniform manifold approximation and projection (UMAP) dimension reduction with the top 20 principal components were performed. A nearest-neighbor graph using the 20 dimensions of the PCA reduction was calculated using FindNeighbors, followed by clustering using FindClusters. Where fewer dimensions of the PCA were informative, as indicated by an "elbow" in a plot of standard deviation vs principal components, we adjusted the FindNeighbours function accordingly. We used the doHeatmap and FeaturePlot functions to visualize the results of the clustering procedure and z-scaled expression per cell. Marker genes for each cluster were identified by the FindAllMarkers function using the Wilcoxon rank sum test. Over-represented KEGG pathways and GO biological process categories for the 100 markers with the greatest log2FC (FDR < 0.05) in each cluster were identified by enrichKEGG and enrichGO functions in the clusterProfiler package (v4.2)[59]. Cell cycles were classified with the cyclone function from the scran R package[60] using the default parameters and the mouse gene pairs included in the package.

### Statistical analysis

Statistical analyses were performed using the Student's *t* test, non-parametric Wilcoxon test and Peason *r* correlation using GraphPad software. Data is shown as mean ± SD. Symbols represent: *$p < 0.05$, **$p < 0.01$, ***$p < 0.001$, and ****$p < 0.0001$.

### Reporting summary

Further information on research design is available in the Nature Research Reporting Summary linked to this article.

## Data availability

The raw data and processed scRNA-seq and bulk data generated in this study have been deposited in the Gene Expression Omnibus (GEO) database under the accession number GSE206977, which can be assessed with no restriction. The source data underlying Figs. 2b, 3d-f, 4a-c, 4e-f, 5c-g, 6c, 6e-f, 7d, 7g, and Supplementary Figs. 1a, 2e, 3a-b, 3d-f, 3h, 4a-c, 6a-d are in the Source Data file provided with this paper. Source data are provided with this paper.

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

## Acknowledgements

This work was supported by grant NHLBI R01 DK097837 and HL141379 (N.C.), R01 HL145170 and a MCBC Pilot Project from City of Hope (Z.B.C.) and CIRM EDU4-12772 (HK). We thank the shared facility resources at City of Hope: Animal Resource Core, Flow Cytometry and Sorting Core, Microscopy and Imaging Core and Integrative Genomics Core supported by the NCI/NIH grant P30CA033572. We thank Dr. Ian Talisman for editing and Dr. Nora Heisterkamp for critical review of the manuscript.

## Author contributions

Y-W.K. and N.C. conceived the study, interpreted results, wrote, reviewed, and approved the final version of the manuscript. Y.-W.K. designed and performed experiments and analyzed results. G.Z. performed complementary experiments, confirmed independently the culture methods and reviewed the manuscript. H.K. conducted experiments for the revision. S.B. and D.O. analyzed RNA-seq data. Y.F. performed preliminary experiments. C-YK generated the *Tie2-CreERT2* construct for the mouse strain. M.N., A.B., and R.R. analyzed and quantified Matrigel networks. Y.L. and Z.B. prepared RNAs for RNA-seq and reviewed the manuscript. Y.Z. provided *Tie2-CreERT2;Rosa26-tdTomato* mice and preliminary data. A.C. interpreted results and reviewed the manuscript.

## Competing interests

The authors declare no competing interests.
