## [Peer Review File · Nature Communications]

Integration of single-cell transcriptomes and biological function reveals distinct behavioral patterns in bone marrow endotheliumREVIEWER COMMENTS

Reviewer #1 (Remarks to the Author):

Kim et al describe a new method for expanding BM sinusoidal endothelial cells ex vivo, thoroughly evaluated with a combination of molecular and cellular approaches. They specifically demonstrate that BM SECs can be expanded by ~ 50-fold over 14 days with minimal loss of cell-lineage identity if cultured with total BM cells. After 14 days, cell passaging results in loss of BM SECs. Using cell sorted populations they reveal a significant role for macrophages in supporting BM SECs, and a possibly inhibitory effect of BM mesenchymal cells in maintenance of BM SECs ex-vivo. In contrast, arterial-like ECs expand several hundred fold and do not require the presence of macrophages or other BM cells. These conclusions are corroborated by bulk and single cell RNA sequencing analyses of EC populations from freshly sorted BM cells, cultured BM cells, and of ECs from other organs (e.g. lung). Overall, this is an interesting set of studies that advances the field and is suitable for publication. A few minor concerns were noted and are listed below, which will render this study ready for publication after being addressed.

- 1- It's unfortunate that Fig S1e does not include liver and spleen. Without this information, authors should tone down statements about the "uniqueness" of BM SECs.
- 2- The strategy for identifying mesenchymal cells is sub-optimal and so the use of expressions such as "lineage identity" (page 6) should be avoided.
- 3- The data indicating a role for macrophages is interesting but not impressive. Emcn+ ECs expanded ~ 48 fold when cultured with total BM, whereas macrophages barely made BM Emcn+ SECs expand (less than 2-fold; unclear if this is even expansion or just improved survival). Authors should rephrase their conclusions to reflect the very mild role played by macrophages in these culture conditions.
- 4- Related to the last point, authors may want to test if adding MCSF to the 2-cell culture assay (Emcn+ ECs and Macs) improves the expansion of BM SECs.
- 5- It would be interesting to test if the MC inhibitory effect requires cell contact or is via soluble factor(s).
- 6- Page 8, please define MSC and include details on how these cells were isolated.
- 7- Insufficient details on culture medium used, cytokines/growth factors added, concentrations, and their commercial sources, catalog numbers, etc. For a "methods" study this section requires extended details.
- 8- Methods describe experiments using Osx-cre mice but it's unclear where data from these mice are presented. Can authors clarify this?
- 9- Details are also needed for how MSC cells were isolated and identified.

Reviewer #2 (Remarks to the Author):

In their manuscript with the title "Integration of single-cell transcriptomes and biological function reveals distinct behavioral patterns in bone marrow endothelial cells", Kim et al. have isolated and characterized genetically labelled endothelial cells (ECs) from murine marrow (BM). They report that freshly isolated BMECs represent two different populations, which differ in the expression of the marker Endomucin (Encn) and exhibit distinct behaviors in culture. The authors also show that co-culture with other BM-derived cells has a big impact on BMECs. BMEC growth and expansion is propagated by hematopoietic cells/macrophages, whereas mesenchymal stromal cells (MSCs), which strongly expand in culture, inhibit endothelial cell growth. A final part of the study focuses on the properties of the two BMEC subpopulations. The authors propose that the Emcn- ECs are arterial-like and express stalk cell markers in culture, whereas the Emcn+ population is thought to represent sinusoidal ECs and shows more tip cell-like gene expression after culture. Overall, this is an interesting study providing more insight into the heterogeneity and functional

specialization of BMECs, but, as outlined further below, some of the findings are puzzling and their conceptual relevance for BMEC behavior in vivo remains unclear.

The most critical and really fundamental open question, in my view, is the disconnect between the observations and the function of the bone vasculature in vivo. *Emcn+* ECs exhibit a venous/sinusoidal-like phenotype (page 5), but they also express tip cell markers (page 9). There are two issues here. First, the tip/stalk concept is very good for the description of EC behavior within a growing vessel sprout in the angiogenic vasculature of certain tissues (like retina, brain and many tumors). In contrast, the meaning is much less clear in adult, homeostatic vessel beds and, in particular, in the BM sinusoidal vasculature. The latter shows, to my knowledge, very little or no sprouting behavior and vessel architecture is fundamentally different from sprouting capillary beds. So what would the expression of tip cell-like genes actually mean in terms of biological function? In Fig. 7h, it is notable that a substantial fraction (about 50%) of tip cell-like genes shows low expression in *Emcn+* ECs in vivo (fresh) and is substantially upregulated in culture. This should be considered in the analysis of the data and might substantially alter the interpretation of findings.

There is a similar disconnect for the *Emcn-* population, which, according to the manuscript, is artery-like but expresses stalk cell-like markers. This raises the same questions regarding biological relevance as for the *Emcn+* population mentioned above. What conclusions can be drawn about the biology of these cells and what does gene expression tell us about function? Why can artery-like BMECs grow in culture over several passages and show a high proliferative potential even though arteries display generally very limited proliferation?

It should be also considered that there is substantial overlap between gene expression in tip cells and arterial ECs in other tissues. At the moment, it is unclear whether this could potentially represent a difference between arteries in bone and elsewhere. Alternatively, can it be ruled out that the initial classification of *Emcn-* and *Emcn+* cells is arbitrary and that these cells represent other vessel subtypes in vivo? For example, ref. 23 in the manuscript refers to a non-sinusoidal population with high *Emcn* expression and high expression of Notch pathway genes. This would be also consistent with the presence of two distinct *Emcn+* populations in the scRNA-seq data shown in the current study.

The authors mention in the Discussion that *Emcn-* ECs might represent progenitors, which would substantially alter the interpretation of central data in the manuscript. Again, it is unclear how findings in culture correspond to bone vasculature in vivo.

Another issue that needs clarification are the differences and similarities in gene expression between the EC populations in vivo (freshly isolated) and in culture. The authors state that heterogeneity and identity are largely preserved in culture, but the heat maps provided in Fig. 7g and h indicate very substantial changes in gene expression. Some genes, such as *Ltbp4* and *CD34* in *Emcn-* ECs or *Notch4* and *Apln* in *Emcn+* cells, flip completely and transcripts for all genes with very high expression in vivo appear substantially reduced in culture. One would also expect strong changes because of the very different oxygen levels in vivo (hypoxic bone marrow) and in vitro. The authors need to provide a more careful analysis of their gene expression data and go beyond the superficial statement that gene expression profiles are "remarkably similar". In reality, there are big differences – also according to the UMAP plot in Fig. 7e – and it is important to outline transcriptional changes because they are relevant for the interpretation of the data.

It is very interesting that macrophages promote EC growth in mixed cultures, whereas MSCs have the opposite effect. Unfortunately, there is no explanation for this finding at a mechanistic level and it would be very important to identify candidate factors that mediate the communication with BMECs in mixed culture.

It is stated that BMECs show "robust tube formation in Matrigel" but the low-resolution overview images provided only show cord-like networks. Evidence for the formation of endothelial tubes with a distinct lumen is lacking.

The authors repeatedly state that they have used a new (Tie-CreERT2; Rosa26-tdTomato) reporter mouse for the genetic labelling of ECs in vivo, but, in fact, this particular combination of alleles has been previously published in 2019 (Hochstetler et al., Exp Hematol). This previous study also talks about a "novel inducible" Tie2-CreERT2 mouse model and refers to a published abstract from 2012 for further details. In my view, "new" is a misnomer here and not justified. In addition, I have noted that there is no publication describing the actual generation of these mice so that it remains unknown which approach and construct have been used. The lack of such fundamental and essential information is not acceptable and needs to be rectified.

Fig. S2a and S2b: The magnification is too low. Arteries should be indicated and, preferentially, labeled by staining with an arterial marker. Absence of Emcn signal alone is insufficient.

The term "colony-forming" cells for Emcn- Sca-1^{hi} ECs might be confused with clonal colony formation, which is not shown. The authors should consider "cluster-forming cells" or other alternative terms.

Response to the reviewers

We thank the reviewers for their comments and their thoughtful input on our manuscript, which we believe has improved as result of this process. We have enclosed below our response to the reviewers' specific comments, as a point-by-point response to their major critiques (in blue italics), and the revised manuscript. We have included clarifications and performed additional experiments to address the reviewers' concerns.

As a result of this revision, we added panels to Figures S1a, S1b and S2b. We also added new Figures: 5g, S1h, S2e, S3c, S4a, S4c, S5b, S5c and S8a. We edited the manuscript accordingly and included the recommended points in the revised Discussion. Major changes in the manuscript are tracked in dark red.

We are hopeful that the changes introduced into this manuscript will satisfy the reviewers' concerns.

Reviewer #1:

1- It's unfortunate that Fig S1e does not include liver and spleen. Without this information, authors should tone down statements about the "uniqueness" of BM SECS.

- We agree and have added another set of PO cultures of various organs that includes liver and spleen. We revised Figure S1e, now Figure S1c, to include snapshot images of PO cultures of liver and spleen at day 14. 2D networks can be seen only in PO Whole Bone Marrow (WBM) culture. Thus, we believe that we can still include statements regarding the "uniqueness" of BMECs.

In addition, to highlight the results for all organs, we have rearranged Figures 1 and S1: the previous Figure S1a and S1b, are now Figure 1a and 1b.

2- The strategy for identifying mesenchymal cells is sub-optimal and so the use of expressions such as "lineage identity" (page 6) should be avoided.

- We have performed a more complete characterization of this population by adding the expression of additional markers known to be highly expressed on mesenchymal cells: CD90, CD29, Sca-1, and CD44 expression in the absence of CD45, and CD11b. These data are now presented in Figure S3c. This more complete characterization supports our use of the expression "lineage identity" in the text.

3- The data indicating a role for macrophages is interesting but not impressive. Emcn+ ECs expanded ~ 48 fold when cultured with total BM, whereas macrophages barely made BM Emcn+ SECS expand (less than 2-fold; unclear of this is even expansion or just improved survival). Authors should re-phrase their conclusions to reflect the very mild role played by macrophages in these culture conditions.

- We realized that these results were not clearly described. In Figures 5a and 5c, we show that in standard WBM culture by day 14 (containing 1x number of macrophages) Emcn+ BMECs expand 48-fold compared to day 0. When we added twice as many macrophages (2x) to the standard WBM culture, the growth of Emcn+ cells increased a further ~2-fold, from 48-fold to ~96-fold. The increase in BMEC growth in the presence of macrophages is also shown as an increase in the tdT+ fluorescence area (reflecting the number and size of colonies) in the new Figure S4a. Furthermore, sorted BMECs cultured by themselves had limited growth compared to BMEC cultured in WBM (Figure 2b) but were capable of greater expansion when macrophages were added to the sorted BMECs (Figure S4b) and we found a significant correlation between decreased numbers of Emcn+ and decreased numbers of macrophages from P0 to P2. Last, addition of M-CSF to the P0 WBM culture increased Emcn+ and Emcn- BMEC output.

Collectively, these data suggest that macrophages play a positive role in supporting EC proliferation and survival. We have worded this more clearly in the Results section and figure legend.

4- Related to the last point, authors may want to test if adding M-CSF to the 2-cell culture assay (Emcn+ ECs and Macs) improves the expansion of BM SECS.

- We thank the reviewer for proposing this experiment, which tests whether endogenous increase of macrophages also induces expansion of BMECs. We have performed this experiment, and indeed addition of M-CSF induces expansion of SECS. This data is shown in Figure 5g.

5- It would be interesting to test if the MC inhibitory effect requires cell contact or is via soluble factor(s).

- *We conducted additional experiments evaluating the growth of EC in the presence of MC or macrophages in the absence or presence of transwell inserts. Results are shown in Figure S4c. Data suggest that cell contact is critical for the MC-mediated effect on BMEC, whereas the effect of macrophages is still present with transwell inserts, indicating the importance of soluble factors.*

6- Page 8, please define MSC and include details on how these cells were isolated.

- *Figure 7a shows our comparison of bulk-RNA-seq from BM P0 Emcn+, BM P0 Emcn- and Lung Emcn- to determine how these populations differ from each other and ultimately how all differ from BM MSCs. To be confident of the mesenchymal identity, we used Osx-Cre;Rosa26-tdTomato mice, because the Osx promoter is well known to be expressed in multipotent BM mesenchymal progenitor cells. We harvested and cultured Osx-Cre;Rosa26-tdTomato mice BM cells in Mesencult medium (Stem Cell Technologies). At day 14, P0 cultures were collected and DAPI-CD45-tdT+ cells (MSCs) were sorted and used for bulk RNA-seq. We apologize for the lack of clarity in the original description and have added these details in the figure legend.*

7- Insufficient details on culture medium used, cytokines/growth factors added, concentrations, and their commercial sources, catalog numbers, etc. For a “methods” study this section requires extended details.

- *We apologize for the oversight. We have added detailed information on EGM-2 medium in the Methods section.*

8- Methods describe experiments using Osx-cre mice but it's unclear where data from these mice are presented. Can authors clarify this?

- *Osx-Cre;Rosa26-tdTomato mice were used to derive MSCs to conduct the bulk-RNA-seq comparison shown in Figure 7a. We have clarified this in the Figure 7a legend and Methods section.*

9- Details are also needed for how MSC cells were isolated and identified. *Please see points 6 and 8.*

Reviewer #2 (Remarks to the Author):

In their manuscript with the title “Integration of single-cell transcriptomes and biological function reveals distinct behavioral patterns in bone marrow endothelial cells”, Kim et al. have isolated and characterized genetically labelled endothelial cells (ECs) from murine marrow (BM). They report that freshly isolated BMECs represent two different populations, which differ in the expression of the marker Endomucin (Encn) and exhibit distinct behaviors in culture. The authors also show that co-culture with other BM-derived cells has a big impact on BMECs. BMEC growth and expansion is propagated by hematopoietic cells/macrophages, whereas mesenchymal stromal cells (MSCs), which strongly expand in culture, inhibit endothelial cell growth. A final part of the study focuses on the properties of the two BMEC subpopulations. The authors propose that the Emcn- ECs are arterial-like and express stalk cell markers in culture, whereas the Emcn+ population is thought to represent sinusoidal ECs and shows more tip cell-like gene expression after culture.

Overall, this is an interesting study providing more insight into the heterogeneity and functional specialization of BMECs, but, as outlined further below, some of the findings are puzzling and their conceptual relevance for BMEC behavior in vivo remains unclear.

1- The most critical and really fundamental open question, in my view, is the disconnect between the observations and the function of the bone vasculature in vivo. Emcn+ ECs exhibit a venous/sinusoidal-like phenotype (page 5), but they also express tip cell markers (page 9). There are two issues here. First, the tip/stalk concept is very good for the description of EC behavior within a growing vessel sprout in the angiogenic vasculature of certain tissues (like retina, brain and many tumors).

In contrast, the meaning is much less clear in adult, homeostatic vessel beds and, in particular, in the BM sinusoidal vasculature. The latter shows, to my knowledge, very little or no sprouting behavior and vessel architecture is fundamentally different from sprouting capillary beds. So what would the expression of tip cell-like genes actually mean in terms of biological function?

In Fig. 7h, it is notable that a substantial fraction (about 50%) of tip cell-like genes shows low expression in *Emcn*⁺ ECs *in vivo* (fresh) and is substantially upregulated in culture. This should be considered in the analysis of the data and might substantially alter the interpretation of findings.

There is a similar disconnect for the *Emcn*⁻ population, which, according to the manuscript, is artery-like but expresses stalk cell-like markers. This raises the same questions regarding biological relevance as for the *Emcn*⁺ population mentioned above. What conclusions can be drawn about the biology of these cells and what does gene expression tell us about function?

- The reviewer raises important questions, but we and the field do not have all the answers. Our approach has unveiled previously unknown aspects of vascular biology. Thus, a major value of this work is in making new observations that raise new questions for study and providing a platform that can be helpful to address them.

*Our scRNA-seq analysis shows that adult, homeostatic vessels do express genes classified as “tip” or “stalk” genes. Other analyses have revealed similar expression of some of these genes (i.e. Tikhonova AN, et al. Nature **569**, 222-228, 2019). but the connection with tip/stalk classification was not made. We examined a more extensive list of the known “tip” and “stalk” genes and found a preserved tip/stalk signature in fresh BMECs that have sinusoidal-like or arterial-like identities, respectively. One speculation is that adult vessels at homeostasis maintain a baseline expression of a restricted group of “tip/stalk” genes – perhaps memory of a sprouting occurred – that can be poised to be induced to higher expression when a situation necessitating regeneration occurs. This condition is mimicked by the “artificial” angiogenic conditions of *in vitro* culture, in which we observed a stronger “tip” signature with re-activation of additional “tip” genes not expressed in fresh cells. A similar explanation could be proposed for the “stalk genes” expressed in the arterial-like BMEC. It is possible that this population represents a physiological reservoir of regenerative cells in the BM, which are quiescent and not activated during steady-state conditions, but are activated by “angiogenic” *in vitro* culture conditions. We have integrated this point into the revised discussion.*

2 - Why can artery-like BMECs grow in culture over several passages and show a high proliferative potential even though arteries display generally very limited proliferation?

*- In vivo, arterial-like endothelial cells are part of vessel structures maintained at low levels of proliferation and regeneration at steady-state; however, we know that these cells can be induced to proliferate and re-activate morphogenesis in response to stress or damage. It is possible that the presence of supra-physiologic angiogenic factors in *in vitro* culture conditions triggers a similar regenerative response. We anticipate that our platform will help to elucidate these types of questions in future studies.*

3 - It should be also considered that there is substantial overlap between gene expression in tip cells and arterial ECs in other tissues. At the moment, it is unclear whether this could potentially represent a difference between arteries in bone and elsewhere. Alternatively, can it be ruled out that the initial classification of *Emcn*⁻ and *Emcn*⁺ cells is arbitrary and that these cells represent other vessel subtypes *in vivo*? For example, ref. 23 in the manuscript refers to a non-sinusoidal population with high *Emcn* expression and high expression of Notch pathway genes. This would be also consistent with the presence of two distinct *Emcn*⁺ populations in the scRNA-seq data shown in the current study.

*- We agree with the reviewer that an arbitrary component is present in all types of classifications and that all classifications have limits. In our study, we observed that the *Emcn*⁺ and *Emcn*⁻ distinction more accurately identifies cells that have sinusoidal vs. arterial characteristics (both *in vitro* and *in vivo*) than does the classical *Sca1*^{high} *Sca1*^{low} classification commonly adopted (1,2) or PDPN (3) expression. Still, like in the other approaches, we recognize that there is some promiscuity of signatures, that classification is just a tool to guide us and can be changed, and there is more need to be understood. Indeed, we have also observed that a third group of endothelial cells, defined by high *Emcn*, express both arterial and sinusoidal genes. These cells are very rare and are lost *in vitro*. It is possible that they are a transient population or a progenitor population or they may be type H EC (4). Their rarity and that they cannot be preserved *in vitro* using current methods, makes studying them challenging. We are planning to characterize them in the near future.*

We have also given substantial thought to the fact that some genes, like those in the Notch pathway (e.g., Notch1, Notch4, Dll4) are considered “arterial” genes. These genes are also classified as “tip” genes and the fact that we found they were more highly expressed in association with sinusoidal genes and in “sinusoidal”-like EC may appear as a contradiction. Although we found this result initially confusing, we would argue that the classification of Notch1, Notch4, Dll4 as “arterial” genes is arbitrary and may not be entirely correct. These genes play a pivotal role in arterial “specification” during

development, but it is not clear to what extent they are essential for maintaining “arterial” identity in adult tissue. In the adult tissue and at steady state, one could think of these genes not in terms of identity, but in terms of function—that is being expressed in cells that are more actively remodeling, such as sinusoids.

1. Yuya Kuniski et al. Arteriolar niches maintain haematopoietic stem cell quiescence. *Nature*. 2013 Oct 31; 502(7473). PMID: 24107994. PMCID: PMC3821873
2. Cesar Nombela-Arrieta et al. Quantitative imaging of haematopoietic stem and progenitor cell localization and hypoxic status in the bone marrow microenvironment. *Nat Cell Biol*. 2013 May; 15(5). PMID: 23624405. PMCID: PMC4156024
3. Chunliang Xu et al. Stem cell factor is selectively secreted by arterial endothelial cells in bone marrow. *Nature Commun*. 2018 June 22; 9(1):2449. PMID: 29934585 PMCID: PMC6015052
4. Anjali P Kusumbe et al. Coupling of angiogenesis and osteogenesis by a specific vessel subtype in bone. *Nature*. 2014 Mar 20;507(7492). PMID: 24646994 PMCID: PMC4943525

4 - The authors mention in the Discussion that *Emcn*- ECs might represent progenitors, which would substantially alter the interpretation of central data in the manuscript.

*- This sentence in the discussion was intended as speculation. We mentioned that *Emcn*- ECs could be progenitors because they have high proliferative potential. We agree with the reviewer that this statement can raise confusion, especially because we do not provide evidence that these cells can differentiate into a different cell type. Therefore, we respectfully accepted the reviewer’s suggestion and have removed the sentence.*

5 - Another issue that needs clarification are the differences and similarities in gene expression between the EC populations in vivo (freshly isolated) and in culture. The authors state that heterogeneity and identity are largely preserved in culture, but the heat maps provided in Fig. 7g and h indicate very substantial changes in gene expression. Some genes, such as *Ltbp4* and *CD34* in *Emcn*- ECs or *Notch4* and *Apln* in *Emcn*+ cells, flip completely and transcripts for all genes with very high expression in vivo appear substantially reduced in culture. One would also expect strong changes because of the very different oxygen levels in vivo (hypoxic bone marrow) and in vitro. The authors need to provide a more careful analysis of their gene expression data and go beyond the superficial statement that gene expression profiles are “remarkably similar”. In reality, there are big differences – also according to the UMAP plot in Fig. 7e - and it is important to outline transcriptional changes because they are relevant for the interpretation of the data.

- We appreciate the reviewer’s comments, which have encouraged us to better clarify the “similarity” relationship between freshly isolated and in-culture cells. We have added a more detailed explanation in the Results section for clarity.

*The similarity we described in gene profile between in vivo and in culture cells, in both the *Emcn*+ and *Emcn*- groups, is demonstrated not only through visual inspection of the UMAP plots, but also through the objective, mathematical method used to determine the dendrogram structure shown above the heatmap (now Figure 8g and h and Figure S8a). The results show the linkage between freshly isolated and cultured cells of the respective groups (*Emcn*- and *Emcn*+). The analysis was carried out by first building a distance matrix between the subpopulation’s transcriptome set using Euclidian distance, and the Ward method to perform the hierarchical clusters and build the dendrogram. Moreover, the above-described pattern persists regardless of the number of genes we include in the analysis. In Figures 8g and 8h of the manuscript, we have reported type-specific genes (arterial & sinusoidal and tip & stalk). As a new analysis, in supplementary figure S8a, we show an analysis of the most variable genes (those in the top 5 percentile of variability) to correct for possible bias in the choice of gene subset. This data-driven approach confirmed the observed pattern previously obtained and strengthens the assertion regarding the strong transcriptomic similarity between in vivo and in culture cells for each population. It is important to note that the populations in the two conditions show great similarities in the transcriptome, but they are not identical.*

*The reviewer’s observation that a few genes, in the type-specific subsets, flipped expression between the in vivo and in vitro conditions, is, of course, of biological interest and can be explained, as noted by the reviewer, by the different environmental conditions between in vivo and in vitro. However, this does not undermine the idea of similarity and difference between the different subgroups as revealed by the mathematical analysis of the high dimensional space of the different subsets of genes. This is true both in the UMAP projection (for which we used all genes), in which the cell subclusters *Emcn*+ (or *Emcn*-) colocalized and were clearly separated by *Emcn* expression, and in the dendrogram topology shown above the heatmap, which also showed the aforementioned topology.*

We recognize that while we sought to emphasize the similarity of BMEC cells in vivo and in vitro in the previous version, we overlooked discussing the genes that changed expression in response to culture conditions. We have added such discussion in the revised Results section.

6 - It is very interesting that macrophages promote EC growth in mixed cultures, whereas MSCs have the opposite effect. Unfortunately, there is no explanation for this finding at a mechanistic level and it would be very important to identify candidate factors that mediate the communication with BMECs in mixed culture.

- We agree with the reviewer that this is an interesting finding and points to endothelial cells requiring a specific “niche” for their optimal growth. We also note that identifying and defining the specific molecular mechanisms underlying the macrophage- and MSC-mediated effects could take may be a year or so, and goes beyond the scope of this report. However, we did conduct some experiments that provide additional information. 1) We have corroborated the role of macrophages by showing that addition of M-CSF to the culture medium leads to increased numbers of BMECs (Figure 5g). 2) We have performed co-culture experiments of BMEC, macrophages and mesenchymal cells using transwell inserts. These experiments show that the mesenchymal cells’ inhibitory effect is mediated via cell-to-cell contact whereas the effect of macrophages seems to be mediated mostly by soluble factors (Figure S4b). We hope that the reviewer finds this information to be a sufficient step forward to narrow down the mechanisms. We intend to use our platform to reveal additional mechanistic information in further studies.

7- It is stated that BMECs show “robust tube formation in Matrigel” but the low-resolution overview images provided only show cord-like networks. Evidence for the formation of endothelial tubes with a distinct lumen is lacking.

- We have added high-resolution and high magnification images to Figure S5b. The reviewer is correct; the structures are cord-like networks. In our previous version of the paper, we indicated these structures as “tubes” to be consistent with the literature, which has used the “tubes” definition to refer cord-like structures in hundreds of studies. Below we provide 3 papers as examples of the many that show and quantitate tube-assay results without documenting lumen (1-3). The images shown in these papers are very similar to ours.

That said, we agree with the reviewer that the use of “tube” without demonstrating formation of a distinct lumen is not correct. However, we understand and note that the broad use of “tubes” is in recognition of cells that have engaged in the morphogenesis process within the limitations of current “tube” assays. This terminology has become the norm, despite a published consensus guideline recently reported in “Angiogenesis” (4).

In the revised version we changed the term to “cords” and “cord-networks”. To avoid confusion for the readers, we include a figure and explanation in supplementary figure S5c.

1. DeCicco-Skinner et al. Endothelial Cell Tube Formation Assay for the *In Vitro* Study of Angiogenesis. *J Vis Exp.* 2014; (91): 51312. PMC4540586 PMID: 25225985
2. Arnaoutova I, Kleinman HK. In vitro angiogenesis: endothelial cell tube formation on gelled basement membrane extract. *Nat Protoc.* 2010 Apr;5(4):628-35. PMID: 20224563
3. ML Ponce. Tube formation: an in vitro matrigel angiogenesis assay. *Angiogenesis Protocols*, 2009 Springer.
4. Patrycja Nowak-Sliwinska et al. Consensus guidelines for the use and interpretation of angiogenesis assays. *Angiogenesis* 2018 Aug; 21(3): 425–532. PMC6237663

8 - The authors repeatedly state that they have used a new (Tie-CreERT2; Rosa26-tdTomato) reporter mouse for the genetic labelling of ECs in vivo, but, in fact, this particular combination of alleles has been previously published in 2019 (Hochstetler et al., *Exp Hematol*). This previous study also talks about a “novel inducible” Tie2-CreERT2 mouse model and refers to a published abstract from 2012 for further details. In my view, “new” is a misnomer here and not justified. In addition, I have noted that there is no publication describing the actual generation of these mice so that it remains unknown which approach and construct have been used. The lack of such fundamental and essential information is not acceptable and needs to be rectified.

*- We agree with this comment. This mouse line was generated and used by our collaborator Dr. Yi Zheng (Cincinnati Children’s Hospital), reported by his group in an abstract in *Blood* (2012), and subsequently used in his study by Hochstetler et al. (*Exp Hematol* 2019). Technically, it is not “new”. We used “new” broadly, with the intention of indicating that the model has been published only recently and used in only one publication by the group that developed it. We have revised*

our wording to avoid using “new”. We did not include the description of this mouse line in our manuscript because there was intention from Dr. Zheng to publish description of the strain independently. Since the development of mouse model has not yet been reported, our collaborators agreed to provide a detailed description of generation of this line in this manuscript. It is now included in the Methods section.

9- Fig. S2a and S2b: The magnification is too low. Arteries should be indicated and, preferentially, labeled by staining with an arterial marker. Absence of Emcn signal alone is insufficient.

- We agree that the magnification is too low to see a single blood vessel in detail. However, we intentionally chose to show the low magnification image because we wanted to show the whole vascular structure in each organ, without focusing on a single blood vessel. Emcn has been validated as being expressed only in venous and capillary ECs (1,2) and not in arterial ECs, thus arteries can be distinguished as Emcn-tdT+ blood vessels. To be more accurate, as the reviewer recommended, we have added anti- α SMA labeling to validate the arterial phenotype (see Figure S2b). We have also added white arrowheads in each image to mark arteries.

1. Suzanne Marie Morgan et al. Biochemical characterization and molecular cloning of a novel endothelial-specific sialomucin. Blood. 1999 Jan 1;93(1). PMID: 9864158
2. Anjali P Kusumbe et al. Coupling of angiogenesis and osteogenesis by a specific vessel subtype in bone. Nature. 2014 Mar 20;507(7492). PMID: 24646994 PMCID: PMC4943525

10 -The term “colony-forming” cells for Emcn- Sca-11hi ECs might be confused with clonal colony formation, which is not shown. The authors should consider “cluster-forming cells” or other alternative terms.

- We acknowledge the lack of clarity for this point. In our study, we consistently observed that single cells generate a colony. Sometimes, growth and extension of colonies leads to them merging with each other, thus appearing as larger colonies. We have added snapshot images from a representative colony in Figure S1h and a movie (Movie 2) to illustrate this process. These images were taken by monitoring the exact same spot in the culture dish of P0 WBM culture at days 2, 5, 7, 9, and 14. Reviewing these images shows clonal expansion.

REVIEWERS' COMMENTS

Reviewer #2 (Remarks to the Author):

The authors have addressed all my questions and have modified the text accordingly. I have no further comments or queries.

I think this is a nice study that very prove very useful to the community.